# Excitatory neurons are more disinhibited than inhibitory neurons by chloride dysregulation in the spinal dorsal horn

Kwan Yeop Lee[1,2,3], Stéphanie Ratté[1,2,3], Steven A Prescott[1,2,3]*

[1]Neurosciences and Mental Health, The Hospital for Sick Children, Toronto, Canada; [2]Department of Physiology, University of Toronto, Toronto, Canada; [3]Institute of Biomaterials and Biomedical Engineering, University of Toronto, Toronto, Canada

**Abstract** Neuropathic pain is a debilitating condition caused by the abnormal processing of somatosensory input. Synaptic inhibition in the spinal dorsal horn plays a key role in that processing. Mechanical allodynia – the misperception of light touch as painful – occurs when inhibition is compromised. Disinhibition is due primarily to chloride dysregulation caused by hypofunction of the potassium-chloride co-transporter KCC2. Here we show, in rats, that excitatory neurons are disproportionately affected. This is not because chloride is differentially dysregulated in excitatory and inhibitory neurons, but, rather, because excitatory neurons rely more heavily on inhibition to counterbalance strong excitation. Receptive fields in both cell types have a center-surround organization but disinhibition unmasks more excitatory input to excitatory neurons. Differences in intrinsic excitability also affect how chloride dysregulation affects spiking. These results deepen understanding of how excitation and inhibition are normally balanced in the spinal dorsal horn, and how their imbalance disrupts somatosensory processing.

*For correspondence:
steve.prescott@sickkids.ca

Competing interests: The authors declare that no competing interests exist.

## Introduction

Neuropathic pain results from damage to or dysfunction of the nervous system. It affects ~10% of the population (*van Hecke et al., 2014*) and is notoriously difficult to treat (*Woolf, 2010*). Hypersensitivity to tactile stimulation is a troubling feature of such pain. This so-called mechanical allodynia can be acutely reproduced by blocking synaptic inhibition at the spinal level (*Miraucourt et al., 2009*; *Sivilotti and Woolf, 1994*; *Sorkin and Puig, 1996*; *Sorkin et al., 1998*; *Yaksh, 1989*). Numerous other studies have shown that synaptic inhibition in the spinal dorsal horn is reduced after nerve injury (for reviews, see *Prescott, 2015*; *Price et al., 2009*). Disinhibition can result from reduced activation of GABA$_A$ or glycine receptors, or from reduced current flow through activated receptors. The former has several possible causes (*Zeilhofer et al., 2012*) but the latter stems uniquely from dysregulation of intracellular chloride due to KCC2 hypofunction (*Coull et al., 2003*). Enhancing KCC2 function reverses injury-induced allodynia (*Gagnon et al., 2013*; *Lavertu et al., 2014*; *Mapplebeck et al., 2019*), thus demonstrating that chloride dysregulation contributes significantly to injury-induced disinhibition.

Knowing the molecular mechanism through which disinhibition occurs is important for devising therapeutic interventions, but the effects of disinhibition must also be considered at the circuit level to explain how sensory processing is disrupted (or conversely, to infer how disinhibition arises based on observable/reportable sensory changes that reflect circuit function). Indeed, selective disinhibition of inhibitory or excitatory dorsal horn neurons would likely yield opposite effects. Chloride dysregulation has been shown to occur in dorsal horn neurons with different spiking patterns

(*Coull et al., 2003*), but circuit operation (and sensory processing) depends on how much inhibition each cell type normally experiences and how much is lost under pathological conditions.

Disinhibition is known to unmask low-threshold input to projection neurons that normally respond exclusively to noxious input (*Baba et al., 2003*; *Cheng et al., 2017*; *Keller et al., 2007*; *Lavertu et al., 2014*; *Lu et al., 2013*; *Miraucourt et al., 2007*; *Miraucourt et al., 2009*; *Torsney and MacDermott, 2006*). Low-threshold input is conveyed polysynaptically to projection neurons via excitatory interneurons (for reviews, see *Duan et al., 2018*; *Peirs and Seal, 2016*; *Prescott et al., 2014*; *Takazawa and MacDermott, 2010b*). The disproportionate effect of disinhibition on low-threshold input (resulting in allodynia) suggests that excitatory interneurons are disinhibited; greater effects on noxious input (resulting in hyperalgesia) would be expected if disinhibition occurred selectively in projection neurons, that is after the convergence of low- and high-threshold inputs. Separate work indicates that excitatory interneurons are necessary for mechanical allodynia (*Cheng et al., 2017*; *Duan et al., 2014*; *Hu et al., 2006*; *Malmberg et al., 1997*; *Wang et al., 2013*). Inhibitory neurons also receive inhibitory input (*Takazawa and MacDermott, 2010a*; *Zheng et al., 2010*), and could, therefore, also become disinhibited, but whether the resultant increase in GABAergic/glycinergic transmission enhances inhibition depends on the degree of chloride dysregulation in postsynaptic neurons (*Doyon et al., 2011*; *Prescott et al., 2006*). Thus, despite many advances, many questions remain about the balance of excitatory and inhibitory input to different dorsal horn neurons and how this E-I balance becomes pathologically altered.

Beyond 'gating' sensory input in the spinal dorsal horn (*Melzack and Wall, 1965*), synaptic inhibition plays an important role in organizing receptive fields (RFs) (*Isaacson and Scanziani, 2011*). As originally described in retinal cells by *Kuffler (1953)*, many neurons have an RF with an excitatory center and inhibitory surround. This includes dorsal horn neurons (*Hillman and Wall, 1969*). A center-surround organization has important implications for sensory processing and for how that processing is disrupted by disinhibition. Notwithstanding other explanations, reduced two-point discrimination in patients with neuropathic pain (*Pleger et al., 2006*) is consistent with increased RF overlap that could occur if surround inhibition is lost and RFs expand. Animal studies have demonstrated that RFs expand after nerve injury (*Behbehani and Dollberg-Stolik, 1994*; *Cumberbatch et al., 1998*; *Devor and Wall, 1981*; *Suzuki et al., 2000*; *Tabo et al., 1999*). This will affect the spatial summation of diffuse (broadly distributed) stimuli – moreso than for punctate (spatially restricted) stimuli – and is liable to affect dynamic mechanical allodynia, which is commonly provoked by clothes touching the skin.

The present study began with an exploration of how chloride dysregulation affects the RFs of superficial dorsal horn neurons. To this end, we blocked spinal KCC2 in vivo while monitoring the responses of neurons to tactile stimulation of the hind paw. Beyond RFs expanding, we found that unmasked excitatory input from the original RF surround became the predominant source of excitation. In vivo recordings also revealed two distinct cell types: units identified by their adapting spike pattern during tactile stimulation correspond to excitatory neurons whereas non-adapting units correspond to inhibitory neurons. In vitro patch clamp recordings showed that chloride regulation did not differ between excitatory and inhibitory neurons, and in vivo application of BDNF verified that both cell types experience chloride dysregulation under pathological conditions. That said, adapting and non-adapting cells responded differently to equivalent chloride dysregulation in part because of differences in intrinsic properties and in part because of differences in E-I balance at the circuit level. Most notably, inhibitory neurons were found to receive weak excitation counterbalanced by weak inhibition, whereas excitatory neurons receive strong excitation counterbalanced by strong inhibition. Because excitatory neurons rely more heavily on inhibition, they are disproportionately affected by disinhibition.

## Results

### Disinhibition causes cutaneous receptive fields to expand

For a RF with center-surround organization, disinhibition is expected to expand the RF by unmasking excitatory input originating from the surround (*Figure 1A*). This will enhance spatial summation, which is critical for diffuse stimuli – like the rubbing of clothes – that typically provoke dynamic allodynia. Compared with the punctate allodynia evoked by stimuli like von Frey hairs, dynamic allodynia

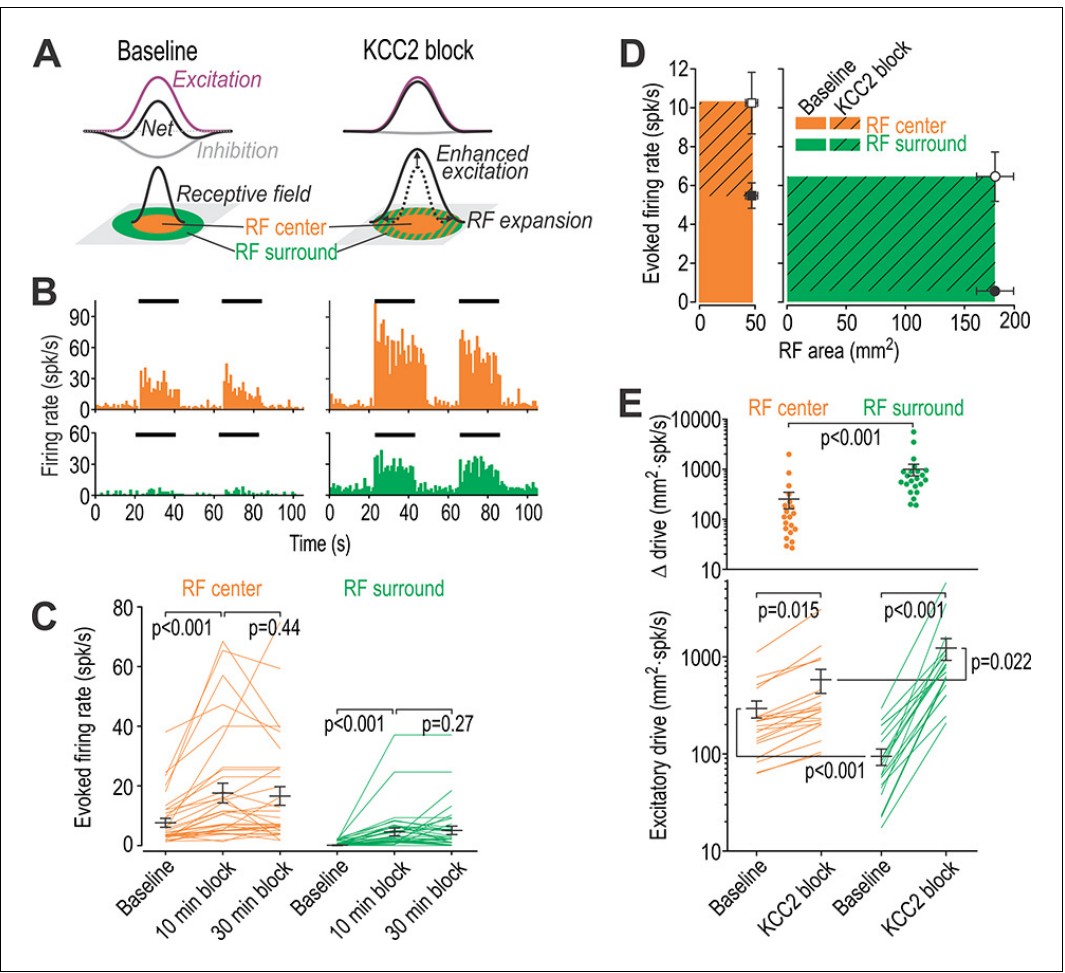

**Figure 1.** Disinhibition causes receptive fields to expand. (**A**) Cartoon depicts how narrowly tuned excitation combined with broadly tuned inhibition produces a receptive field (RF) with an excitatory center and inhibitory surround. (**B**) Firing rate histograms from a typical neuron responding to brush stimulation (black bars) applied to the RF center (orange) or surround (green) before (left) and after (right) blockade of KCC2 by intrathecal DIOA. (**C**) Summary of spike rates evoked by brush stimulation in the RF center or surround 10 and 30 min after application of 100 µM DIOA ($n$ = 10) or 50 µM VU ($n$ = 22) to block KCC2; data for DIOA and VU were pooled based on the absence of any significant differences. KCC2 blockade had a significant effect on brush-evoked firing ($F_{2,62}$ = 17.78, p<0.001, two-way repeated measures ANOVA). Results of Wilcoxon tests are reported on graphs. Firing rates were stably elevated for at least 20 min, thus allowing for other effects of disinhibition (e.g. on RF size) to be measured. (**D**) To quantify the excitatory drive originating from each RF zone, average evoked firing rate (± SEM) was plotted against average surface area (± SEM) of the RF zone in which the stimulus was applied. Each data point demarcates a rectangle (colored) whose area is proportional to excitatory drive from that RF zone. (**E**) By calculating the product of RF zone area and firing rate on a cell-by-cell basis ($n$ = 22), disinhibition was found to significantly increase total drive ($F_{1,21}$ = 139.17, p<0.001; two-way repeated measures ANOVA on log-transformed data) but this depended on the RF zone (interaction, $F_{1,21}$ = 49.81, p<0.001): the RF surround contributed significantly greater drive than the RF center after KCC2 blockade. Results of Student-Newman-Keuls (SNK) tests are reported on graphs.

The online version of this article includes the following source data for figure 1:

**Source data 1.** Numerical values for data plotted in *Figure 1*.

is more troubling for patients (*Hansson, 2003*). Dorsal horn neuron RFs have been reported to change on a short timescale (minutes), consistent with an RF shaped by the balance of excitatory and inhibitory input rather than being 'hard wired' (*Cook et al., 1987*; *Dubuisson et al., 1979*; *Laird and Cervero, 1989*; *McMahon and Wall, 1984*). We therefore set out to determine how the RF size of spinal lamina I-II neurons depends on chloride regulation.

To test the effects of KCC2 blockade on RF size, we used brush stimulation to map the cutaneous RF of dorsal horn neurons recorded in vivo. The RF measured under normal (baseline) conditions was labeled the RF *center*; by definition, areas outside the RF center evoked negligible spiking (*Figure 1B* left). After blockade of KCC2 by intrathecal DIOA, stimulation in the RF center evoked larger responses and stimulation outside the RF center evoked vigorous spiking (*Figure 1B* right). The RF was re-mapped and the additional area was labeled the RF *surround*. KCC2 blockade significantly increased firing evoked by stimulation in the RF center or surround ($Z = 4.79$, p<0.001 and $Z = 4.72$, p<0.001 respectively; Wilcoxon tests) (*Figure 1C*). Firing rate histograms in *Figure 1B* include all spikes but the spontaneous firing rate preceding each stimulus was subtracted from firing during stimulation to focus on evoked firing in *Figure 1C–E*.

Next, we compared the enhanced excitatory drive from the RF center with the unmasked excitatory drive from the RF surround by plotting the average area for each RF zone against the average firing rate evoked by stimulation in that zone. This defines a rectangle whose area is proportional to the total excitatory drive contributed by that RF zone (*Figure 1D*). By multiplying the area of each RF zone with the corresponding firing rate for each unit (i.e. calculating the area of the aforementioned rectangle on a cell-by-cell basis), we found that KCC2 blockade significantly increased drive from both the RF center and surround ($Q_{21} = 3.60$, p=0.015 and $Q_{21} = 18.54$, p<0.001, respectively; post-hoc Student-Newman-Keuls [SNK] tests). Excitatory drive from the RF surround was (by definition) significantly less than from the RF center under baseline conditions ($Q_{21} = 5.91$, p<0.001), but the inverse was true after disinhibition ($Q_{21} = 3.42$, p=0.022) (*Figure 1E*). Hence, after disinhibition caused by KCC2 blockade, neurons were driven more strongly by unmasked excitatory input from the RF surround than by enhanced input from the RF center.

## Units recorded in vivo comprise two distinct groups

Over the course of these experiments we noticed that some units had a monophasic spike waveform whereas others had a biphasic waveform (*Figure 2A*). The former exhibited spike rate adaptation during sustained tactile stimulation with a von Frey (VF) filament whereas the latter did not (*Figure 2B*). The adaptation time constant measured from exponential-rise-to-maximum fits of the cumulative probability distribution of spike times from each unit differed significantly between units classified by their spike waveform ($T_{23} = 8.58$, p<0.001; unpaired *t*-test) (*Figure 2B* inset). The difference in adaptation argues that units with distinct spike waveforms represent distinct cell types rather than, for example, somatic and axonal recording from the same population of neurons. The two cell types are henceforth referred to as *adapting* and *non-adapting*. In subsequent experiments, we found that approximately 80% of units responsive to light tactile stimulation could be readily classified by their adaptation pattern; unclassified units were excluded from subsequent analysis.

Analysis of additional response properties support the classification based on adaptation pattern. The RFs of adapting units were significantly smaller than RFs of non-adapting units at baseline ($U = 17.5$, p=0.013; Mann-Whitney test) (*Figure 2C*). KCC2 blockade enlarged RFs in both adapting units ($Q_{14} = 11.69$, p<0.001; SNK test) and non-adapting units ($Q_6 = 5.94$, p<0.001). Of the units analyzed in *Figure 1E*, 12 were subsequently identified as adapting and four as non-adapting; the disinhibition-mediated increase in drive from the RF surround relative to increased drive from the RF center tended to be greater in adapting units, though the difference did not reach significance ($T_{14} = 1.89$, p=0.080; unpaired *t*-test) (*Figure 2D*).

To assess functional connectivity between primary afferent neurons and each type of spinal unit, we recorded simultaneously from the dorsal root ganglion and spinal cord. Mechanosensitive afferents were subdivided into rapid-adapting (RA) and slow-adapting (SA) (*Johansson and Vallbo, 1979*). We calculated cross-correlograms by measuring the probability of a spinal unit firing at different times relative to each spike in a simultaneously recorded primary afferent with overlapping RF. According to this analysis, adapting units receive tactile input exclusively from RA afferents whereas non-adapting units receive tactile input preferentially from SA afferents (*Figure 2E*). Differences in adaptation during evoked firing may be due to differences in primary afferent input and feedforward inhibition (*Zhang et al., 2018*), but differences in spontaneous spiking (see ) suggest that adapting and non-adapting spinal units also differ in their intrinsic excitability. Indeed, data in *Figure 2E* may reflect differential responsivity of spinal units rather than differential connectivity per se: Neurons operating as coincidence detectors respond preferentially to synchronous inputs (*Ratté et al., 2013*) that are likely to arise from RA afferents whose spikes occur synchronously (*Lankarany et al., 2019*),

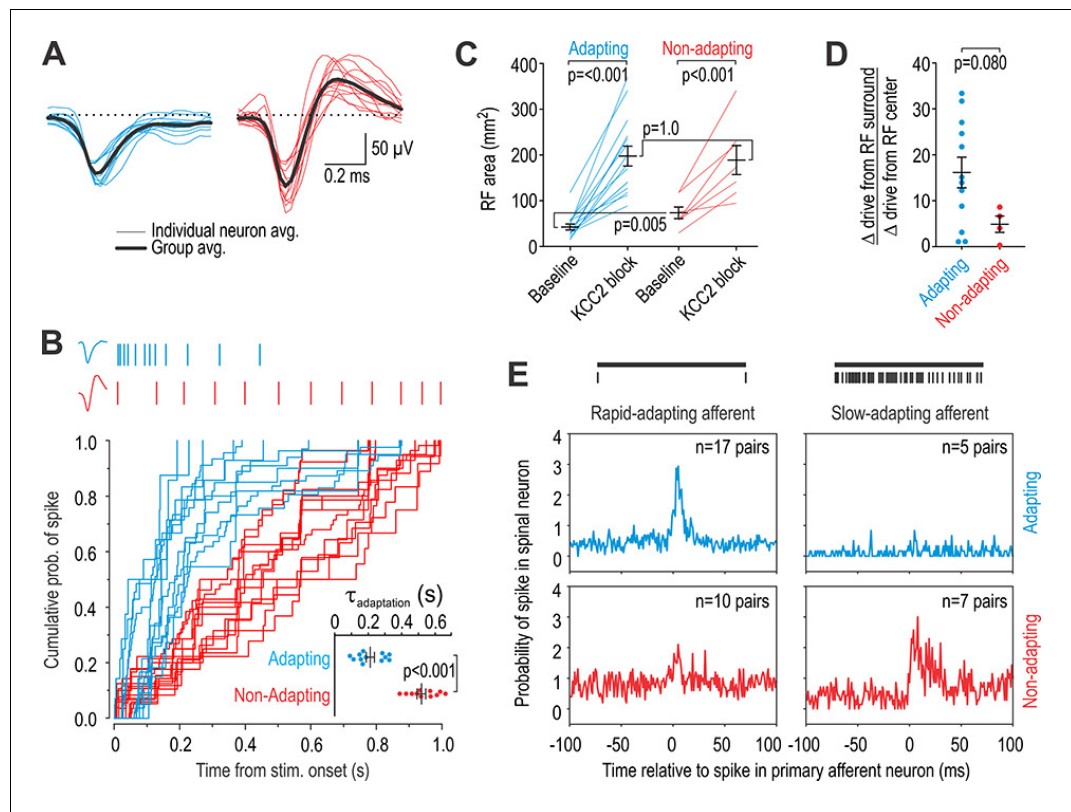

**Figure 2.** Units recorded in vivo can be divided into two groups. (**A**) Spike waveforms were separated according to their monophasic (blue) or biphasic (red) shape. (**B**) Units subdivided by spike waveform also differed in their adaptation pattern, as illustrated by sample rasters showing responses to 1 s-long 10 g von Frey stimulation (top). The cumulative probability of spiking was fit for each unit with the curve $y = 1/(1-e^{-t/\tau})$. The adaptation time constant $\tau$ differed significantly between groups ($T_{23}$ = 8.58, p<0.001; unpaired t-test). Units are henceforth referred to as adapting (blue) or non-adapting (red). (**C**) Adapting units had a significantly smaller RF than non-adapting units at baseline ($Q_{21}$ = 17.5, p=0.013; Mann-Whitney test) but there was no difference in RF size after KCC2 blockade ($Q_{21}$ = 49.0, p=0.83). RF size was significantly affected by KCC2 blockade in both type of units ($F_{1,20}$ = 66.10, p<0.001; two-way repeated measures ANOVA). Results of SNK tests are reported on graphs. (**D**) KCC2 blockade caused a larger, though not quite significant, increase in drive from the RC center relative to the RF surround in adapting units compared with non-adapting units ($T_{14}$ = 1.89, p=0.080; unpaired t-test). (**E**) Cross-correlograms were calculated from all pairs of simultaneously recorded primary afferent neurons and spinal neurons with overlapping RFs. The number of pairs is reported on each panel. Primary afferents were classified as rapid-adapting (RA, left) or slow-adapting (SA, right); rasters show sample responses to 1 s-long 10 g von Frey stimulation (top). Adapting units displayed an increased probability of spiking after RA spikes (top left) but not after SA spikes (top right) whereas non-adapting units displayed a large increase in spike probability after SA spikes (bottom right) and a modest increase after RA spikes (bottom left). In other words, adapting units receive input exclusively from RA afferents whereas non-adapting units receive input preferentially from SA afferents. The online version of this article includes the following source data for figure 2:

**Source data 1.** Numerical values for data plotted in *Figure 2*.

whereas neurons operating as integrators also respond to asynchronous inputs that are likely to arise from SA afferents whose spikes occur asynchronously. Natural tactile stimuli will co-activate both types of spinal neurons, but may do so in different proportions depending on stimulus characteristics. Recent efforts using optogenetics to identify which type of low-threshold mechanoreceptors mediate allodynia after nerve injury are interesting in this regard: *Dhandapani et al. (2018)* showed that activating TrkB-ChR2-positive neurons – which include D-hair (Aδ) and RA afferents – provokes allodynia whereas *Chamessian et al. (2019)* showed that activating Vglut1-ChR2-positive neurons – which include RA and SA afferents – does not. The discrepancy suggests that activation of RA *but not* SA afferents produces allodynia, which implies that spinal circuits implement a NIMPLY logic

gate. This is consistent with our data and supports combinatorial coding over labeled lines (*Ma, 2010*; *Prescott et al., 2014*; *Prescott and Ratté, 2012*). These are important considerations for future studies.

## Adapting and non-adapting units correspond to excitatory and inhibitory neurons, respectively

Next, we asked whether adapting and non-adapting units represent different cell types within the dorsal horn circuit. Since projection neurons are absent in lamina II and represent only ~5% of neurons in lumbar segments of lamina I in rats (*Spike et al., 2003*), the units we recorded were most likely local interneurons, of which approximately 2/3 are excitatory and 1/3 are inhibitory (*Abraira et al., 2017*; *Polgár et al., 2003*; *Prescott and Ratté, 2012*; *Todd, 2010*; *Todd, 2017*). Without transgenic rats to genetically identify neuron types, and rather than switch to mice in which certain experiments (e.g. *Figure 2E* or 8B) would be prohibitively difficult, we undertook a series of tests designed to link $_{adaptation}$ pattern to pharmacologically testable markers and other distinctive properties.

First, we intrathecally applied agonists of receptors expressed selectively by excitatory or inhibitory dorsal horn neurons to test for differential effects in adapting and non-adapting units. Metabotropic glutamate receptor five is expressed almost exclusively on excitatory neurons and its agonist 3,5-dihydroxyphenylglycine (DHPG) has been shown to selectively modulate the excitability of excitatory neurons (*Hu and Gereau, 2011*). Comparing brush-evoked firing before and after intrathecal DHPG revealed that adapting cells were preferentially affected ($\chi^2$ = 9.95, p=0.007) (*Figure 3A*), consistent with adapting cells being excitatory neurons. Spontaneous spiking was unchanged by DHPG, consistent with adapting units not spiking spontaneously under any condition (see Discussion). Moreover, DHPG is known to reduce the A-type potassium current (*Hu and Gereau, 2011*), which, because of its gating properties, predominantly affects the spiking driven by abrupt depolarization (e.g. AMPA receptor-mediated excitation) (*Zhang et al., 2018*).

The somatostatin (SST) receptor $sst_{2a}$ is expressed selectively on inhibitory neurons (*Todd et al., 1998*). Intrathecal application of SST preferentially affected brush-evoked spiking in non-adapting cells ($\chi^2$ = 16.85, p<0.001) (*Figure 3B*), consistent with non-adapting cells being inhibitory neurons. Unlike the majority (55%) of non-adapting units whose evoked firing was increased by SST, evoked firing was reduced in a minority (32%) of adapting units presumably via an indirect circuit-level effect, namely, increased inhibition mediated by $sst_{2a}$-expressing neurons. Non-adapting units also exhibited an increase in spontaneous firing (see *Figure 3B*), which must be a direct effect of SST on those units given the absence of SST-induced spontaneous firing in any other units. Spontaneous spiking has been previously reported in preprodynorphin neurons (*Zhang et al., 2018*), which express the $sst_{2a}$ receptor (*Kardon et al., 2014*). Notably, $sst_{2a}$ receptor activation causes hyperpolarization via activation of an inward rectifying potassium current according to in vitro recordings (*Kim et al., 2002*), contrary to the excitatory effect that we observed in vivo. The reason for the inverted effect is unclear, but one possibility is that application of a high concentration of SST caused receptor internalization (*Stroh et al., 2000*) and paradoxically reduced normal $sst_{2a}$ signaling; indeed, early electrophysiological studies of SST reported conflicting excitatory and inhibitory effects (e.g. *Mueller et al., 1986*).

*Figure 3C* summarizes the depth of each unit from the dorsal surface of the spinal cord. The distributions of adapting and non-adapting units were significantly different (*D* = 0.56, p<0.001; Kolmogorov-Smirnov test) with an average depth (± SD) of 156 ± 72 μm and 234 ± 57 μm, respectively (*Figure 3C* left), which is consistent with the increasing proportion of inhibitory neurons between lamina I to III (*Polgár et al., 2003*). The relative numbers of adapting and non-adapting units (*n* = 64 and 41, respectively) did not differ significantly ($\chi^2$ = 0.74, p=0.39) from the overall 2:1 ratio of excitatory to inhibitory neurons in the superficial dorsal horn (see above), though one must consider that we recorded selectively from neurons responsive to innocuous tactile stimulation. We identified an additional 27 units that responded to tactile stimulation only after KCC2 blockade (*Figure 3C* right). These units presumably receive low-threshold input via polysynaptic pathways normally blocked by inhibition. Of these units, 22 were adapting and five were non-adapting, consistent with the emerging picture that polysynaptic pathways comprising excitatory interneurons relay low-threshold inputs to superficial projection neurons (see Introduction).

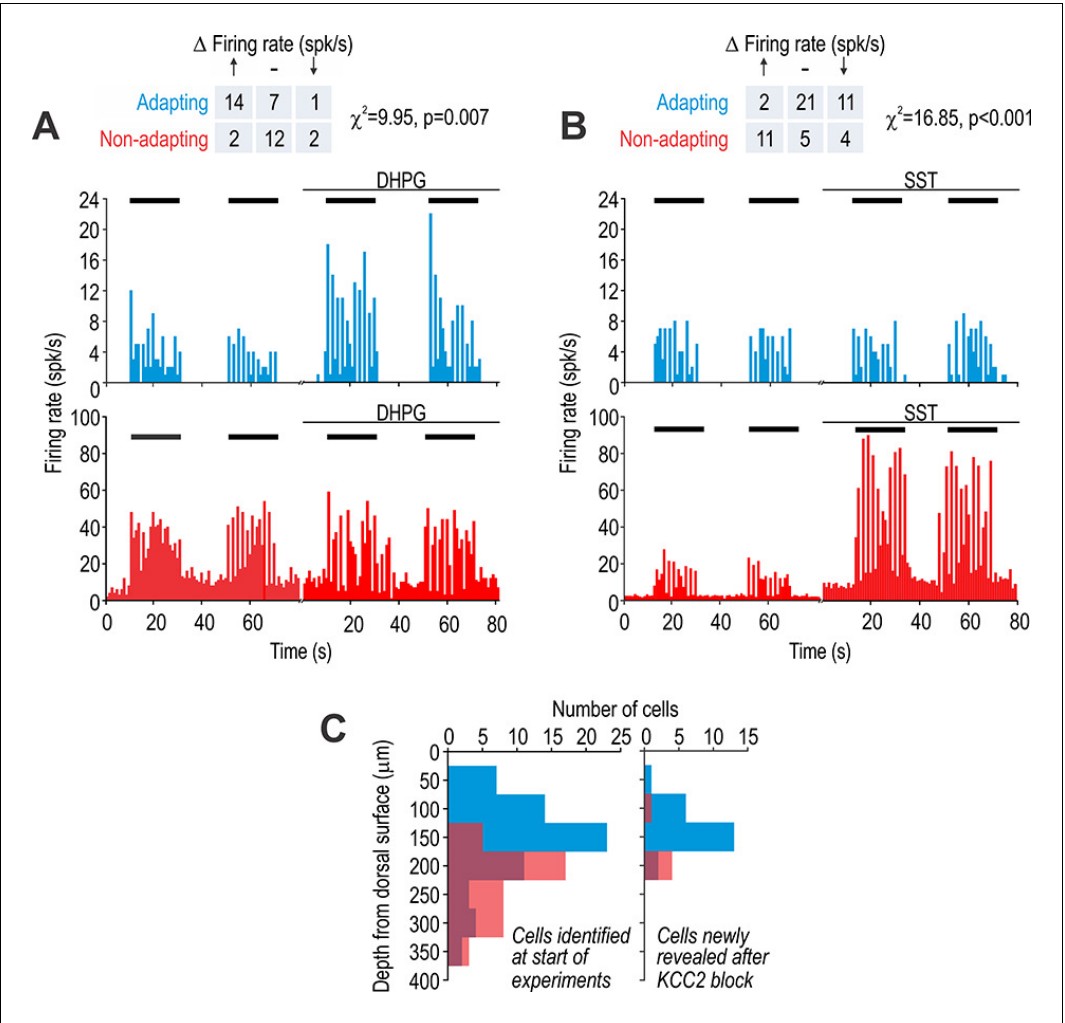

**Figure 3.** Adapting and non-adapting units correspond to excitatory and inhibitory neurons, respectively. (**A**) Sample firing rate histograms from a typical adapting unit (blue) and non-adapting unit (red) during brush stimulation (bars) before and after intrathecal application of 200 μM 3,5-dihydroxyphenylglycine (DHPG). Adapting units were disproportionately affected ($\chi^2$ = 9.95, p=0.007), consistent with them being excitatory neurons, which selectively express metabotropic glutamate receptor 5. (**B**) Sample firing rate histograms before and after intrathecal application of 50 μM somatostatin (SST). Non-adapting units were disproportionately affected ($\chi^2$ = 16.85, p=0.001), consistent with them being inhibitory neurons, which selectively express the $sst_{2a}$ receptor. Some adapting units exhibited reduced spiking (see Results). Spontaneous firing was increased by SST selectively in non-adapting units. (**C**) Distribution of recording depths for units responsive to light touch at baseline (left) and for units that became responsive after KCC2 blockade (right).

The online version of this article includes the following source data for figure 3:

**Source data 1.** Numerical values for data plotted in *Figure 3*.

## Spiking in adapting and non-adapting units is differentially affected by KCC2 blockade

To explore whether putative excitatory and inhibitory neurons are differentially affected by chloride dysregulation, we measured evoked firing before and after KCC2 blockade. *Figure 4A* shows sample input-output (*i-o*) curves for representative adapting and non-adapting units activated by von Frey stimulation of their RF surround. KCC2 blockade significantly increased *i-o* curve slope in adapting and non-adapting units (Z = 3.82, p<0.001 and Z = 3.30, p<0.001, respectively; Wilcoxon tests) (*Figure 4B*). The *y*-intercept was not analyzed since the spontaneous firing rate was subtracted to isolate the evoked response, but separate analysis showed that spontaneous firing was significantly

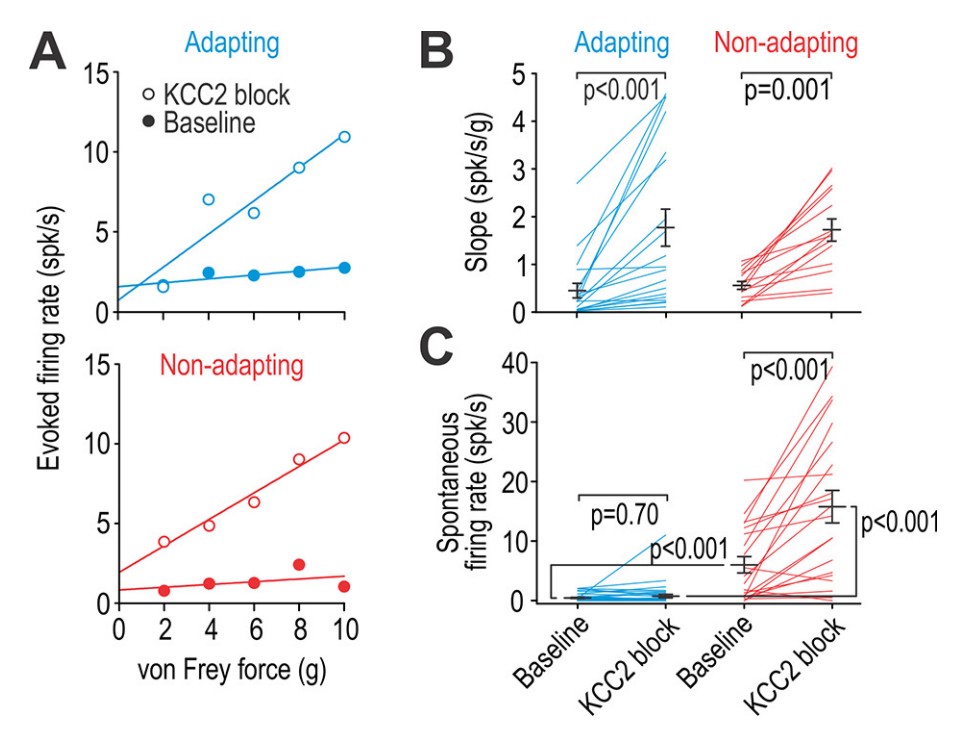

**Figure 4.** Adapting and non-adapting units are differentially affected by KCC2 blockade. (**A**) Input-output (i–o) curves show firing rate evoked by increasing force applied by von Frey hairs to the RF surround in a typical adapting unit (blue) and non-adapting unit (red) before and after blockade of KCC2. (**B**) KCC2 blockade had a significant effect on slope ($F_{1,31}$ = 35.68, p<0.001; two-way repeated measures ANOVA). That effect did not differ between cell types (interaction, $F_{1,31}$ = 0.15, p=0.71) although the largest increases were observed in adapting units. Results of Wilcoxon tests are reported on graphs. (**C**) KCC2 blockade also had a significant effect on spontaneous firing ($F_{1,57}$ = 58.60, p<0.001; two-way repeated measures ANOVA) but that effect differed between cell types ($F_{1,57}$ = 33.79, p<0.001): non-adapting units experienced a significant increase in spontaneous firing ($Z_{19}$ = 3.58, p<0.001; Wilcoxon test) whereas adapting units did not ($Z_{38}$ = 0.39, p=0.70). Spontaneous firing differed significantly between cell types at baseline ($U_{58}$ = 116.5, p<0.001; Mann-Whitney test) and after KCC2 blockade ($U_{58}$ = 55.5, p<0.001). See also **Figure 4—figure supplement 1** for the effect of glycine receptor blockade on spontaneous firing.

The online version of this article includes the following source data and figure supplement(s) for figure 4:

**Source data 1.** Numerical values for data plotted in **Figure 4**.

**Figure supplement 1.** Intrathecal strychnine (50 μM) had no significant effect on the rate of spontaneous firing in non-adapting units ($T_9$ = 1.65, p=0.13; paired t-test).

---

greater in non-adapting units than in adapting units at baseline ($U$ = 116.5, p<0.001; Mann-Whitney test), and that KCC2 blockade significantly increased spontaneous firing in non-adapting units ($Z$ = 3.58, p<0.001; Wilcoxon test) but not in adapting units ($Z$ = 0.39, p=0.70) (**Figure 4C**). Recall that spontaneous firing in non-adapting units was also increased by SST (see **Figure 3B**), yet other manipulations like glycine receptor blockade did not affect spontaneous firing in non-adapting units ($T_9$ = 1.654, p=0.13; paired t-test) (see **Figure 4—figure supplement 1**). The latter observation argues that spontaneous firing is not a consequence of circuit-level disinhibition, but is instead most likely due to the depolarization caused by manipulations like KCC2 blockade. Indeed, inhibitory neurons experience strong tonic GABAergic or glycinergic input (**Gradwell et al., 2017**; **Takazawa and MacDermott, 2010a**), meaning a depolarizing shift in chloride reversal potential will cause depolarization even in the absence of stimulus-evoked synaptic input, and tonic-spiking inhibitory neurons respond to subtle depolarization with sustained spiking because of their voltage-gated channels (**Prescott and De Koninck, 2005**; **Ratté et al., 2014**). To summarize, KCC2 blockade affected

touch-evoked spiking in both adapting and non-adapting units but the selective modulation of spontaneous spiking in the latter likely reflects the intrinsic excitability of that cell type.

## Excitatory and inhibitory neurons are differentially affected by equivalent chloride dysregulation

Differential effects of KCC2 blockade on spiking in adapting and non-adapting units could be due to differences in chloride regulation (KCC2 expression) and/or differences in intrinsic excitability that influence how chloride dysregulation affects spiking. To test the former, we conducted whole-cell patch clamp recordings in a spinal slice preparation (see Materials and methods). Specifically, we identified neurons as excitatory or inhibitory based on their spiking pattern recorded in current clamp (see below) and, after loading the neuron with high chloride, we measured chloride reversal potential ($E_{Cl}$) before and after blocking KCC2 based on voltage clamp responses to puffed GABA (*Figure 5A*). Changes in $E_{Cl}$ measured under a defined chloride load is a more sensitive measure of chloride extrusion capacity than perforated patch measurements of $E_{Cl}$ in low chloride-load conditions (*Doyon et al., 2016*). Results revealed no difference in chloride extrusion capacity between excitatory and inhibitory neurons ($T_{12} = 0.030$, p=0.98, unpaired $t$-test) (*Figure 5B*), thus excluding differences in chloride regulation as the basis for the differentially modulated spiking reported in *Figure 4C*.

For all in vitro experiments, excitatory and inhibitory neurons were identified by their spiking pattern during sustained somatic current injection (see traces on *Figure 5B*). The A-type potassium current, which is associated with delayed- and single-spiking (*Balachandar and Prescott, 2018*), is selectively expressed in the spinal dorsal horn by excitatory neurons (*Hu et al., 2006*) whereas tonic-spiking is observed almost exclusively in inhibitory neurons (*Yasaka et al., 2010*) (for reviews, see *Prescott and Ratté, 2012*; *Todd, 2010*; *Todd, 2017*). This is consistent with excitatory neurons genetically identified by somatostatin expression (most of which exhibit delayed-, single-, or burst-spiking) and inhibitory neurons genetically identified by dynorphin expression (most of which exhibit tonic- or burst-spiking) (*Duan et al., 2014*). Thus, although some spiking patterns (e.g. bursting) are not predictive of cell type, other patterns are (see also *Hughes et al., 2012*). Accordingly, we tested single- or delayed-spiking excitatory neurons and tonic-spiking inhibitory neurons; neurons with other spiking patterns or whose classification was unclear were excluded. Given these inclusion criteria, data in *Figure 5* do not rule out differences in chloride extrusion within certain subpopulations of neurons, but they do rule out a systematic difference between the most common excitatory and inhibitory neurons.

To test whether differences in intrinsic excitability influence how chloride dysregulation affects spiking, we used dynamic clamp to simulate equivalent virtual disinhibition in different cell types. All neurons received virtual excitatory and inhibitory synaptic inputs modeled as Ornstein-Uhlenbeck processes (*Figure 6A*). Firing rate was measured for different average excitatory conductance ($g_{exc0}$). The average inhibitory conductance ($g_{inh0}$) was co-varied with $g_{exc0}$ according to α, where α = ½ represents stimulation inside the RF whereas α = 2 represents stimulation in the RF surround (*Figure 6B*). This parameterization assumes a center-surround organization formed by the combination of narrowly tuned excitatory input and broadly tuned inhibitory input (see Introduction). Chloride dysregulation was modeled as a shift in the anion reversal potential ($E_{inh}$) from −70 mV to −45 mV, where −45 mV represents the largest shift in $E_{inh}$ observed after nerve injury (*Coull et al., 2003*). We used $E_{inh} = −45$ mV to simulate complete blockade of KCC2; partial downregulation of KCC2 after nerve injury would produce a subtler shift.

*Figure 6C* shows the average i-o curves for excitatory neurons (top) and inhibitory neurons (bottom) tested with α = 2. Both types of neurons responded to virtual chloride dysregulation with an increase in firing rate gain (slope) but only inhibitory neurons exhibited a pronounced shift in their average i-o curve. To quantify these changes, we fitted data from each neuron individually and compared the slope and y-intercept between cell types. Disinhibition significantly increased the slope in both excitatory and inhibitory neurons ($Q_5 = 6.62$, p<0.001 and $Q_8 = 3.75$, p=0.020, respectively; SNK tests) (*Figure 6D* top), but increased the y-intercept only in inhibitory neurons ($Q_8 = 8.09$, p<0.001 vs $Q_5 = 0.35$, p=0.81 in excitatory neurons) (*Figure 6D* bottom). The increase in slope tended to be larger for excitatory neurons but the interaction between $E_{inh}$ and cell type did not reach significance ($F_{1,13} = 3.81$, p=0.073). Similar trends were observed for α = ½ (see *Figure 6—figure supplement 1*) though effects of disinhibition were weaker, consistent with weaker inhibition

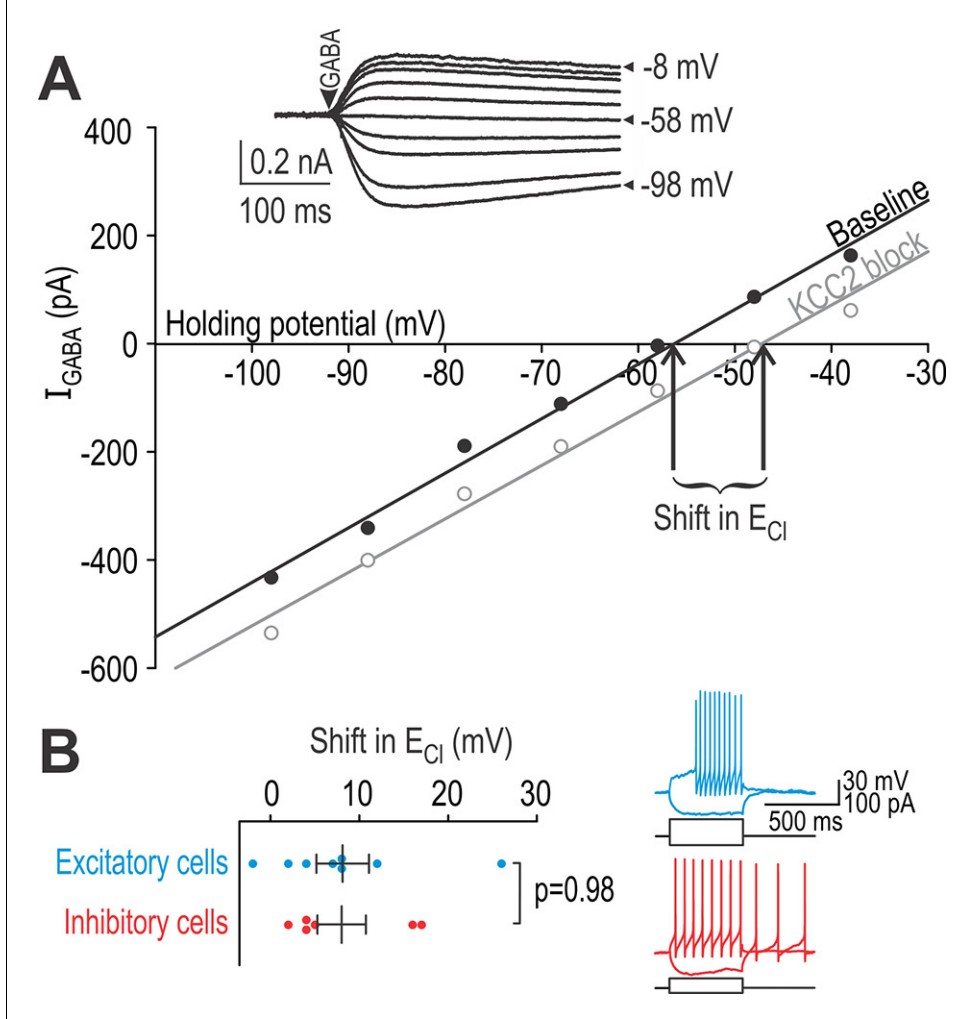

**Figure 5.** Excitatory and inhibitory neurons have equivalent chloride extrusion capacity. (A) Responses to 5–10 ms-long puffs of 1 mM GABA (arrowhead) were measured in voltage clamp at different holding potentials (inset) in neurons patched with a high-chloride (25 mM) pipette solution. Chloride reversal potential $E_{Cl}$, which corresponds to where the *I-V* curve intersects the *x*-axis, was measured before (black) and after (gray) blockade of KCC2 with 15 μM bath-applied VU. The shift in $E_{Cl}$ reflects chloride extrusion capacity, which is not equivalent to the change in $E_{Cl}$ caused by KCC2 blockade under natural conditions (i.e. without intracellular chloride loading). (B) Neurons identified as excitatory or inhibitory based on spiking pattern (inset) displayed no difference in chloride extrusion capacity ($T_{12} = 0.030$, p=0.98; unpaired *T*-test).

The online version of this article includes the following source data for figure 5:

**Source data 1.** Numerical values for data plotted in *Figure 5*.

relative to excitation in the RF center. Indeed, firing rate gain under normal conditions ($E_{inh} = -70$ mV) was significantly less for α = 2 than for α = ½ ($Q_5 = 8.67$, p<0.001 and $Q_5 = 7.45$, p<0.001 for excitatory and inhibitory neurons, respectively; SNK tests), consistent with past work on inhibition-mediated gain control (*Prescott and De Koninck, 2003*; *Silver, 2010*).

*Figure 6E* summarizes the effects of virtual disinhibition in vitro (left panels) for comparison with effects of KCC2 blockade in vivo (right panels). Whereas the firing rate gain of all cell types was significantly increased by disinhibition, only inhibitory neurons (in vitro) and non-adapting units (in vivo) exhibited a significant change in spontaneous spiking. Notably, the relative increase in spontaneous spiking caused by disinhibition is similar between in vitro and in vivo conditions, but the absolute increase is greater in the latter. Also, the differential increase in firing rate gain between excitatory and inhibitory neurons tends to be greater for in vitro experiments than observed in vivo. These

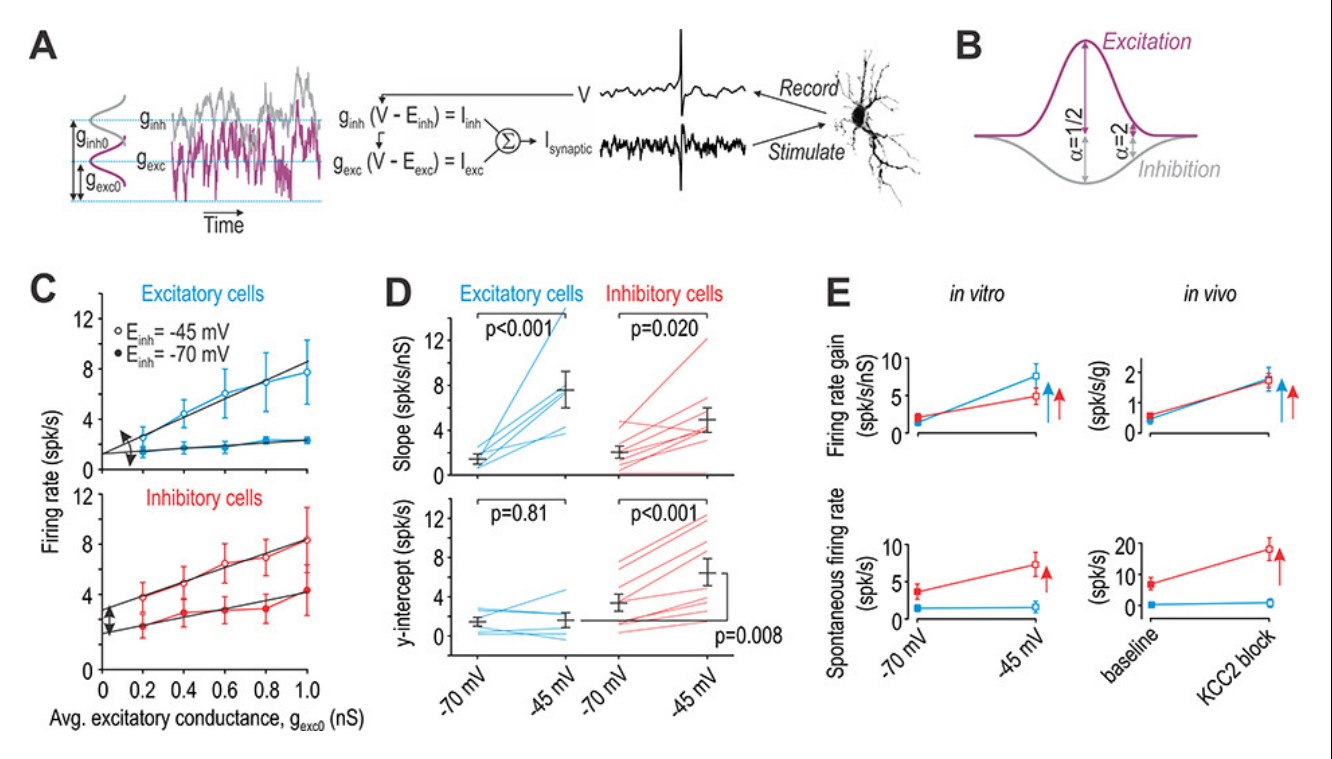

**Figure 6.** Excitatory and inhibitory neurons are differentially affected by virtual disinhibition. (**A**) Cartoon depicts dynamic clamp experiments: noisy excitatory and inhibitory conductances were generated through separate Ornstein-Uhlenbeck processes, converted to currents based on recorded voltage, and applied to the neuron. Average excitatory conductance, $g_{exc0}$, was systematically varied to plot input-output (i–o) curves. (**B**) Average inhibitory conductance, $g_{inh0}$, was co-varied with $g_{exc0}$ according to $\alpha = g_{inh0}/g_{exc0}$, where $\alpha = 0.5$ for virtual stimulation in the RF center and $\alpha = 2$ for virtual stimulation in the RF surround. (**C**) Average i–o curves ($\pm$ SEM) for excitatory cells (blue, $n = 6$) and inhibitory cells (red, $n = 8$) for $\alpha = 2$; see *Figure 6—figure supplement 1* for results with $\alpha = 0.5$. Each neuron was tested with normal inhibition ($E_{inh} = -70$ mV) and again after virtual disinhibition ($E_{inh} = -45$ mV). Disinhibition affected the i–o curve slope of both cell types but shifted the i–o curve only in inhibitory cells. (**D**) $E_{inh}$ had a significant effect on slope ($F_{1,13} = 28.12$, p<0.001, two-way repeated measures ANOVA). That effect did not differ significantly between cell types (interaction, $F_{1,13} = 3.81$, p=0.073) but there was a trend towards a larger increase in excitatory cells. $E_{inh}$ also had a significant effect on the y-intercept ($F_{1,13} = 14.48$, p=0.002) but only in inhibitory neurons (interaction, $F_{1,13} = 11.75$, p=0.004). Results of SNK tests are reported on graphs. (**E**) Summary of disinhibitory effects in vitro (data from panels C and D) for comparison with disinhibitory effects in vivo (data from *Figure 4*). Data are summarized as mean $\pm$ SEM. Each colored arrow indicates a significant effect of disinhibition (p<0.05). Disinhibition significantly increased firing rate gain in all neuron types but increased spontaneous firing selectively in inhibitory neurons and non-adapting units.

The online version of this article includes the following source data and figure supplement(s) for figure 6:

**Source data 1.** Numerical values for data plotted in *Figure 6*.

**Figure supplement 1.** Effects of virtual disinhibition for $\alpha = 0.5$ (virtual stimulation in RF centre).

differences presumably reflect conditions in vivo that were not recapitulated in our in vitro experiments. Notwithstanding quantitative differences, the qualitative concordance between in vitro and in vivo data supports the interpretation of data in *Figure 3*, namely that adapting units represent excitatory neurons whereas non-adapting units represent inhibitory neurons.

## BDNF induces chloride dysregulation in both cell types

Although excitatory and inhibitory neurons exhibit equivalent chloride extrusion capacity according to *Figure 5*, those results do not address whether KCC2 is equivalently downregulated in each cell type after nerve injury. To address this issue, we tested whether adapting and non-adapting units are comparably affected by brain-derived neurotrophic factor (BDNF), which plays a key role in downregulating KCC2 after nerve injury (*Coull et al., 2005*). Notwithstanding sex differences in neuroimmune signaling (*Sorge et al., 2015*), we recently showed that BNDF is sufficient to downregulate KCC2 and cause allodynia in both male and female rodents (*Mapplebeck et al., 2019*). Rather than compare nerve-injured and control animals, in which other changes induced by nerve injury

might compromise our classification scheme, we compared responses for each unit before and after intrathecal BDNF, and re-tested again after applying acetazolamide (ACTZ), which selectively reverses effects of chloride dysregulation by reducing bicarbonate efflux through $GABA_A$/glycine receptors (*Lee and Prescott, 2015*). Reversal of BDNF effects by ACTZ serves as a control for off-target (non-KCC2-mediated) effects of BDNF.

Intrathecal BDNF significantly increased firing evoked by brush stimulation applied to the RF center of adapting and non-adapting units ($Z = 3.10$, p<0.001 for both cell types; Wilcoxon tests) (*Figure 7A* left). Comparable effects were observed for stimuli applied to the RF surround ($Z \geq 2.93$, p<0.001) (*Figure 7A* right). In all cases, BDNF-mediated enhancement of evoked firing was significantly reversed by subsequent application of ACTZ ($Z \geq 2.85$, p≤0.002). Moreover, BDNF failed to induce spontaneous firing in adapting units ($Z = 0.51$ p=0.64) but significantly increased spontaneous firing in non-adapting units ($Z = 2.52$, p=0.008) (*Figure 7B*). This differential change in spontaneous firing is consistent with the effect of KCC2 blockade (*Figure 4C*), and is unlike the effect of glycine receptor blockade (*Figure 4—figure supplement 1*). The BDNF-mediated enhancement of spontaneous firing was significantly reversed by ACTZ ($Z = 2.52$, p=0.008). These results demonstrate that BDNF causes comparable disinhibition in both cell types, but, like for direct blockade of KCC2, spontaneous firing was increased selectively in non-adapting units.

## Surround inhibition is greater, and disininhibition more consequential, for adapting units

In a final set of experiments, we tested how expansion of the RF affects spatial summation of tactile input. To this end, we compared responses to co-stimulation in the RF center *and* surround with responses to stimulation in the RF center alone. Stimulation of the RF surround is predicted to reduce evoked firing under baseline conditions (ratio <1) and to increase it after KCC2 blockade (ratio >1) (*Figure 8A*). As predicted, stimulation in the RF surround significantly reduced the response to stimulation in the RF center ($T_{22} = -9.26$, p<0.001 and $T_{11} = -2.90$, p=0.014 for adapting and non-adapting units, respectively; one sample *t*-tests on log-transformed data) (*Figure 8B*). Adapting units experienced a significantly larger reduction than non-adapting units ($Q_{33} = 3.61$, p=0.013; SNK test). After KCC2 blockade, the effect of stimulation in the RF surround was inverted and enhanced the response to stimulation in the RF center ($T_{22} = 6.24$, p<0.001 and $T_{11} = 6.06$,

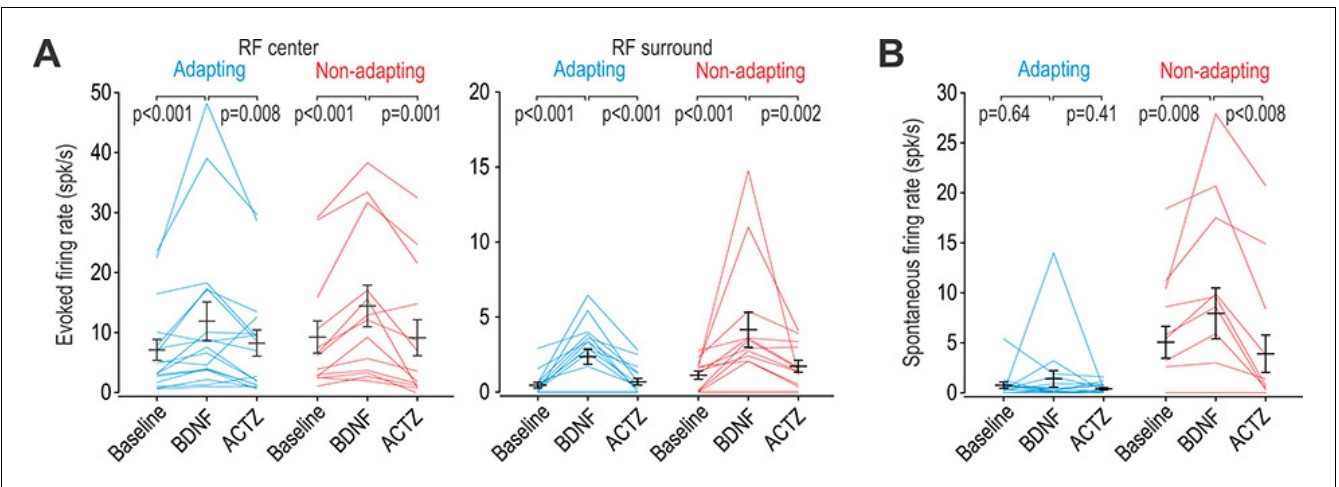

**Figure 7.** BDNF causes disinhibition in both adapting and non-adapting units. (**A**) Intrathecal BDNF (70 µg) and ACTZ (10 mM) had a significant effect on the firing evoked by brush stimulation of the RF center ($F_{2,56} = 15.41$, p<0.001; two-way repeated measures ANOVA) or surround ($F_{2,56} = 17.92$, p<0.001). BDNF significantly increased evoked firing and subsequent application of ACTZ significantly reduced firing in adapting (blue) and non-adapting (red) units. Results of SNK tests are reported on graphs. (**B**) BDNF and ACTZ also had a significant effect on spontaneous firing ($F_{2,56} = 6.76$, p=0.002; two-way repeated measures ANOVA) but that effect differed between cell types (interaction, $F_{1,56} = 8.10$, p=0.008). Only non-adapting units responded to BDNF with significantly increased spontaneous firing, which was entirely reversed by ACTZ. Results of SNK tests are reported on graphs. The online version of this article includes the following source data for figure 7:

**Source data 1.** Numerical values for data plotted in *Figure 7*.

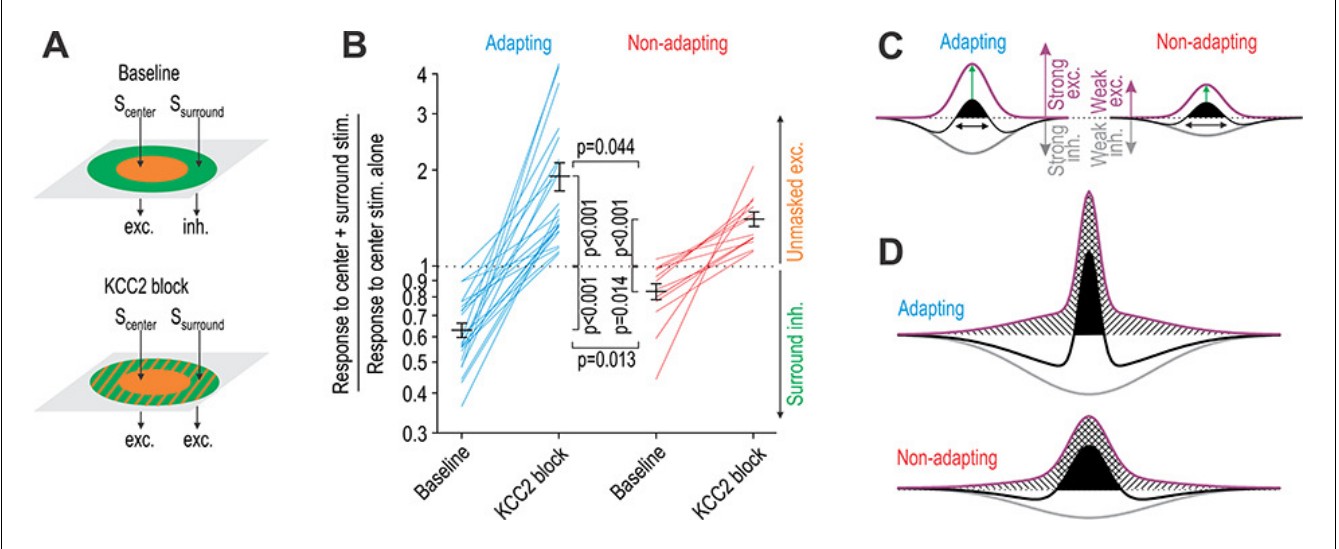

**Figure 8.** KCC2 blockade unmasks significantly more excitation in adapting units than in non-adapting units. (A) Stimulation of the RF center ($S_{center}$) evokes net excitation whereas stimulation of the RF surround ($S_{surround}$) normally evokes net inhibition but evokes excitation after KCC2 blockade. (B) Responses to brush co-stimulation, $S_{center} + S_{surround}$, expressed as a ratio of the response to $S_{center}$ alone. A ratio <1 means that $S_{surround}$ evoked inhibition whereas a ratio >1 means that $S_{surround}$ contributed to excitation. KCC2 blockade had a significant effect on the ratio ($F_{1,33} = 83.93$, p<0.001; two-way repeated measures ANOVA) but that effect differed between cell types (interaction, $F_{1,33} = 8.89$, p=0.005). Specifically, co-stimulation had a significantly stronger inhibitory effect under baseline conditions ($Q_{33} = 3.61$, p=0.013; SNK test) and a significantly stronger excitatory effect after disinhibition ($Q_{33} = 2.92$, p=0.044) in adapting units compared with non-adapting units. (C) A simple model can explain data in panel B: adapting units rely on strong inhibition to counterbalance strong touch-evoked excitation whereas non-adapting units require less inhibition to counterbalance weaker excitation. The RF (black) is defined by the summation of spatially distributed excitatory (purple) and inhibitory (gray) input; RF size is highlighted by black arrows while net excitatory drive is proportional to the area of the black region. However, this version of the model incorrectly predicts that disinhibition (i) increases drive from the RF center more in adapting units than in non-adapting units (green arrows) and (ii) causes a greater increase in drive from the RF center than from the RF surround (see D). (D) The two discrepancies identified in C were resolved through changes to the spatial distribution of excitatory input. Crosshatched and hatched areas show the disinhibition-mediated increase in drive from the RF center and surround, respectively.

The online version of this article includes the following source data for figure 8:

**Source data 1.** Numerical values for data plotted in *Figure 8*.

p<0.001 for adapting and non-adapting units, respectively). Moreover, adapting units experienced a significantly larger enhancement than non-adapting units ($Q_{33} = 2.91$, p=0.044; SNK test). These data suggest that adapting units receive strong excitatory input balanced by strong inhibitory input, whereas non-adapting units receive weaker excitatory input balanced by weaker inhibitory input. Because of their heavier reliance on inhibition to counterbalance strong excitation, adapting units experience greater disinhibition than non-adapting units.

*Figure 8C* depicts the balance of excitatory and inhibitory input described above. The RF at baseline is shown in black; after disinhibition, the RF expands to fill the area under the purple curve. This simple model predicts that adapting units receive more inhibition in the RF center than do inhibitory neurons. To test this, we compared the firing rate evoked by brush stimulation in the RF center before and after KCC2 blockade or BDNF application (using data from *Figures 8B* and *7A*, respectively). The increase in firing caused by KCC2 blockade in adapting units (6.0 ± 1.9 spk/s) and non-adapting units (7.5 ± 3.0 spk/s) was not significantly different ($T_{33} = 0.59$, p=0.56; unpaired t-test). Likewise, the increase in firing caused by BDNF in adapting units (4.8 ± 1.7 spk/s) and non-adapting units (5.2 ± 1.3 spk/s) was not significantly different ($T_{33} = 0.28$, p=0.86; unpaired t-test). These results argue that both cell types experience a similar enhancement of input from the RF center following disinhibition, contrary to what our model predicted. Contrary to data in *Figure 1*, this model also predicted that disinhibition unmasks greater excitation from the RF center than from the RF surround. We therefore asked if and how our initial model could be revised to resolve these discrepancies. Adding a broad component to the excitatory RF for both cell types and slightly narrowing the original excitatory RF for adapting units was sufficient (*Figure 8D*). Consistent with

*Figure 2D*, this revised model correctly predicts that increased input from the RF surround relative to increased input from the RF center is about twice as large for adapting units than for non-adapting units. The broadly tuned excitatory input whose inclusion improved upon the original model may reflect the ungating of polysynaptic circuits, but this and other aspects of the model require additional testing.

## Discussion

Chloride dysregulation has emerged as an important mechanism by which synaptic inhibition can be compromised (*Doyon et al., 2016*). It is now well established in dorsal horn neurons that intracellular chloride levels become elevated after nerve injury, that disrupting chloride regulation (e.g. by KCC2 blockade) produces mechanical allodynia, and that restoring chloride regulation reverses the mechanical allodynia caused by nerve injury (for review, see *Prescott, 2015*). Recent work has verified that these findings apply equally to male and female rodents despite sex-differences in neuroimmune signaling (*Mapplebeck et al., 2019*). Other studies have revealed the diversity of cells types in the dorsal horn and how they connect to form circuits (for reviews, see *Duan et al., 2018*; *Peirs and Seal, 2016*; *Todd, 2017*). But many unknowns remain; for instance, without knowing which cell types experience chloride dysregulation and how their spiking is affected, one cannot predict how chloride dysregulation affects circuit function or, in turn, how altered circuit function produces allodynia. Though informative, manipulations targeting a specific cell type fail to address how changes occurring concurrently in different cell types interact at the circuit level. Moreover, the concept of gating, though invaluable in guiding pain research for over 50 years (*Melzack and Wall, 1965*), is too simplistic to account for complex sensory changes. More detailed investigation of phenomena like spatial and temporal summation is imperative (e.g. *Fitzgerald and Jennings, 1999*), hence the impetus for the current study and its focus on RF structure.

Our results show that dorsal horn neuron RFs are dramatically enlarged by disinhibition, so much so that after KCC2 blockade, neurons receive more excitatory input from the RF surround than they receive from the RF center (*Figure 1*). Though not an initial goal of our study, we were able to separate the majority (~80%) of the units recorded in vivo into two groups that were readily distinguished by their spiking response to tactile stimulation, though several additional features (spike waveform, spontaneous spiking, etc.) also differed between the groups. Pharmacological testing (*Figure 3*) as well as the concordance between in vitro and in vivo testing (*Figure 6E*) argue that adapting and non-adapting units correspond to excitatory and inhibitory neurons, respectively. Both cell types experience chloride dysregulation under pathological conditions, modeled here with BDNF (*Figure 7*), but certain effects manifest differently.

### Effects of disinhibition on spontaneous spiking

KCC2 blockade (as well as BDNF and SST) increased spontaneous spiking selectively in inhibitory neurons (*Figures 3*, *4* and *7*) whereas none of those manipulation, nor DHPG, caused spontaneously spiking in excitatory neurons. Differential modulation of spontaneous spiking was reproduced qualitatively in vitro using dynamic clamp, where virtual disinhibition increased the firing rate gain of both cell types but increased spontaneous spiking (reflected in the *y*-intercept) selectively in inhibitory neurons (*Figure 6*). Differences in intrinsic excitability likely explain the differential effects. First, because of voltage-gated currents, subthreshold depolarization is actively amplified and drives repetitive spiking in tonic-spiking inhibitory neurons but is actively attenuated and fails to drive repetitive spiking in single- and delayed-spiking excitatory neurons (*Prescott and De Koninck, 2005*; *Ratté et al., 2014*). Second, inhibitory neurons are subject to strong tonic inhibition (*Gradwell et al., 2017*; *Takazawa and MacDermott, 2010a*). Glycine receptor blockade did not increase spontaneous spiking in inhibitory neurons (*Figure 4—figure supplement 1*) but a strong tonic chloride conductance would make the resting membrane potential of inhibitory neurons very sensitive to KCC2 blockade. Alone or together, these factors render inhibitory neurons prone to spontaneous spiking when chloride is dysregulated.

Excitatory neurons did not spike spontaneously under normal conditions, and so the absence of (increased) spontaneous spiking after KCC2 blockade might be ascribed to a floor effect. However, excitatory neurons responded vigorously to weak tactile stimulation, which means that they do not operate far from spike threshold, contrary to what a floor effect would imply. Moreover, if non-

adapting units are indeed inhibitory neurons, the continued lack of spontaneous firing in adapting units after KCC2 blockade argues that GABA/glycinergic transmission does not become paradoxically excitatory, which is consistent with past computational work (*Doyon et al., 2011*; *Prescott et al., 2006*) and with past experiments showing that only a small minority of neurons ever exhibit GABA-evoked spikes (*Coull et al., 2003*).

## Effects of disinhibition on evoked spiking

The effect of chloride dysregulation on evoked spiking is more difficult to interpret because of the many additional factors involved, including the type of primary afferent input (*Figure 2E*) and the relative strength of excitation and inhibition triggered by that input, which depends on the circuit. Intrinsic neuronal excitability, reflected in the spiking response to somatic current injection, has also been shown to influence the response to cutaneous stimulation or an in vitro facsimile thereof (*Graham et al., 2007*). At the circuit level, E-I balance depends on where the stimulus occurs relative to the neuron's RF, which changes over time as a dynamic stimulus moves across an RF. The spatio-temporal characteristics of natural tactile stimuli are not readily reproduced with electrical or optogenetic stimulation. By stimulating different parts of the RF alone or in combination (*Figure 8*), our results demonstrate that spatial summation is altered by chloride dysregulation. The consequences of this for sensory processing will depend on the spatial extent of the stimulus on the skin.

A stimulus that extends beyond the RF center and into the RF surround normally engages surround inhibition, causing fewer spikes than if the stimulus was restricted to the RF center. This runs counter to intuition that a broader stimulus will cause greater spatial summation and elicit a bigger response, but it is entirely consistent with experiments in the visual system showing that broader stimuli evoke greater inhibition (*Haider et al., 2010*; *Vinje and Gallant, 2002*). However, when disinhibition occurs, not only is the surround inhibition lost, so too is subliminal excitatory input unmasked. This means that a stimulus extending into the RF surround will engage more total excitation (and drive stronger spiking) than a stimulus limited to the RF center. Furthermore, beyond one neuron being more strongly activated, more neurons will be activated as their expanded RFs encroach on the area of stimulation. For a projection neuron receiving convergent input from excitatory interneurons, these effects of disinhibition compound – more input from each excitatory interneuron and input from more excitatory interneurons – plus reduced inhibition to the projection neuron itself.

Dynamic allodynia provoked by light brushing of the skin is a far greater clinical problem than the allodynia provoked by punctate stimuli or pressure (*Hansson, 2003*). In individuals with neuropathic pain, dynamic allodynia is commonly provoked by clothes moving against the skin, which is precisely the sort of stimulus for which spatial summation is important. That said, preclinical testing of mechanosensitivity relies almost exclusively on von Frey stimulation (*Mogil, 2009*). Moreover, different forms of mechanical allodynia subserved by different primary afferents (*Koltzenburg et al., 1992*; *Ochoa and Yarnitsky, 1993*) and differences in central processing have also been uncovered. *Cheng et al. (2017)* recently found that dynamic allodynia required a set of neurons defined by the co-expression of VGLUT3 and Lbx1, whereas hypersensitivity to von Frey stimulation persisted after those neurons were ablated. Having tested with both brush and von Frey stimuli, our results show that disinhibition caused by chloride dysregulation affects responses to both types of stimuli, which is consistent with most (if not all) neurons being under inhibitory control. Additional work, including behavioral testing, is required to assess how changes in spatial summation due to chloride dysregulation contributes to allodynia. Human psychophysical testing that extends beyond the standard battery of quantitative sensory tests (*Sommer, 2016*) would also be informative. It should be mentioned here that nerve injury or other insults leading to neuropathic pain cause changes beyond chloride dysregulation in spinal neurons, and that the chloride dysregulation triggered by nerve injury may be more modest than modeled here by KCC2 blockade or intrathecal BDNF.

## Effects of disinhibition on E-I balance

By co-stimulating the RF center and surround and comparing the evoked response with responses to stimulation of the RF center alone (*Figure 8*), our study also revealed that E-I balance differs between putative excitatory and inhibitory neurons. Specifically, we found that excitatory neurons receive strong excitation that is counterbalanced by strong inhibition, whereas inhibitory neurons

receive weak excitation that is counterbalanced by weak inhibition. In other words, both cell types appear to receive similar relative levels of excitatory and inhibitory input, but the absolute levels of each input are greater in excitatory neurons. Because excitatory neurons rely on strong inhibition to counterbalance their strong excitatory input, they are more affected than inhibitory neurons by a loss of inhibition. This observation could not have been made through targeted manipulations (e.g. ablation) of a specific cell type, as is now common in many studies (e.g. *Cheng et al., 2017*). That said, our results are fully consistent with excitatory interneurons playing a critical in allodynia. Consistent with earlier work (*Torsney and MacDermott, 2006*), our results argue that disinhibition of those excitatory neurons is key.

The disproportionate disinhibition of excitatory neurons is interesting for two reasons. First, the differential effect is due to circuit organization, that is the amount of excitatory and inhibitory input received by each cell type. Indeed, we did not observe any difference in chloride regulation between cell types (*Figure 5*). Second, sensory processing might not be disrupted if the evoked responses in excitatory and inhibitory neurons were equivalently affected by disinhibition. That said, the disproportionate disinhibition of excitatory neurons is liable to be less evident (or consequential) with punctate stimulation, for which spatial summation is minimal. In contrast to the changes in evoked firing, changes in spontaneous firing may indicate that inhibitory neurons experience stronger tonic inhibition and are disproportionately disinhibited in the absence of stimulation. This may be important for understanding spontaneous pain.

In summary, this study has explored the consequences of chloride dysregulation in superficial dorsal horn neurons. Our results show that both excitatory and inhibitory neurons experience chloride dysregulation but respond differently because of differences in their intrinsic excitability and because of circuit-level differences in E-I balance. Additionally, we demonstrated that dorsal horn neuron RFs have a center-surround organization, and that loss of surround inhibition unmasks substantial excitatory drive, especially in excitatory neurons. Loss of surround inhibition and the unmasking of subliminal excitatory input dramatically alters spatial summation, which may contribute significantly to dynamic allodynia.

## Materials and methods

All experimental procedures were approved by the Animal Care Committee at the Hospital for Sick Children and were performed in accordance to guidelines from the Canadian Council on Animal Care. Experiments were performed on adult male Sprague Dawley rats obtained from Charles River, Montreal. In separate experiments, we have determined that KCC2 is equally important for chloride regulation in males and female rodents, and that KCC2 downregulation contributes equally to neuropathic pain in both sexes (*Mapplebeck et al., 2019*) despite differences in upstream, neuroimmune signaling (*Sorge et al., 2015*).

### Animal preparation for in vivo experiments

Under urethane anesthesia (20% in normal saline; 1.2 g/kg i.p.), a laminectomy was performed to expose L4–S1 segments of the spinal cord. The rat (350–450 g) was placed in a stereotaxic frame and its vertebrae clamped above and below the recording site to immobilize the spinal cord. The left hind paw was immobilized in clay with the plantar surface facing upwards for stimulation. Rectal temperature was maintained at 37°C using a feedback controlled heating pad (TR-200, Fine Science Tools).

### Single unit extracellular recordings

A four-electrode array with a total of 16 recording sites (A4 type, NeuroNexus) was implanted at the L5 spinal level. The array was oriented so that each electrode was at the same mediolateral position. Electrode tips were lowered 328 ± 95 μm (mean ± SD) below the dorsal surface. Since recording sites are spaced at 50 μm intervals up each electrode, we determined on which site a unit was recorded and from this measured the depth of recorded neurons to be <300 μm (see *Figure 3C*), which places them within lamina I or II (*Watson et al., 2008*). Neurons that responded to limb displacement, indicating proprioceptive input, were excluded. The signal was amplified, filtered at 500 Hz – 10 kHz, digitized at 20 kHz with an Omniplex Data Acquisition System (Plexon) and stored with stimulus markers on disk. Single units were isolated using Offline Sorter V3 software (Plexon), and

were analyzed with Neuroexplorer 4 (Plexon). Spike waveforms were identified as monophasic or biphasic based on the absence or presence, respectively, of a negative phase (upward deflection) following the initial positive phase (see *Figure 2A*).

## Drug application in vivo

After the laminectomy, a Vaseline well was built around the spinal cord to target drug delivery to the recording site. In each experiment, 200 µl of saline containing drug was delivered into the well through a small tube (OD: 0.016 inch; Cole-Palmer) inserted intrathecally. To block KCC2, R-(+)-[(2-*n*-butyl-6,7-dichloro-2-cyclopentyl-2,3-dihydro-1-oxo-1H-inden-5-yl)oxy] acetic acid (DIOA; Sigma-Aldrich) and VU0240551 (VU; Tocris Bioscience) were diluted in a stock solution of dimethyl sulphoxide (DMSO) and then diluted in buffered saline for a final concentration of 100 µM for DIOA (*Keller et al., 2007*) and 50 µM for VU (*Lavertu et al., 2014*). Results using DIOA and VU were pooled after determining that there was no difference between them, consistent with past work (*Lee and Prescott, 2015*). 200 µM 3,5-dihydroxyphenylglycine (DHPG; Tocris), 50 µM somatostatin (SST; Abcam), 50 µM strychnine (Abcam) and 70 µg recombinant human brain-derived neurotrophic factor (BDNF; PeproTech) were prepared in saline. Acetazolamide (ACTZ; Sigma-Aldrich) was dissolved in buffered saline with pH 8.2, after which pH was reduced to 7.4 and concentration was adjusted to 10 mM (*Asiedu et al., 2010*).

## Mechanical stimulation and receptive field mapping

Cutaneous receptive fields (RFs) were identified from the spiking evoked by mechanical stimuli applied by brush, blunt probes, or von Frey filaments to the glabrous skin of the left hind paw. By using weak search stimuli, we targeted neurons receiving low-threshold input and avoided causing sensitization. The RF zone eliciting a response under baseline conditions was defined as the RF center. Testing outside the RF center was conducted prior to any pharmacological manipulations (to demonstrate the absence of suprathreshold input) but the RF surround was only mapped after KCC2 blockade. Each stimulus comprised ten 1 s-long applications of the brush or von Frey filament (2, 4, 6, 8, and 10 g) repeated at 2 s intervals. Each stimulus was applied twice onto different locations within each RF zone for each test condition. For experiments in *Figure 8*, two brushes were held together and applied concurrently to two different RF zones.

## In vivo data analysis

To quantify the spiking evoked by mechanical stimulation, we measured the mean firing rate during each stimulus, subtracted the spontaneous firing rate measured from the 10 s epoch preceding each stimulus, and then averaged across the two stimuli for each RF zone and test condition. Input-output (*i-o*) plots produced by varying von Frey force were fitted by linear regression to obtain the slope for each neuron. All data are presented as mean ± SEM.

## Slice preparation for in vitro experiments

Spinal slices were prepared as previously described (*Prescott and De Koninck, 2002*). Briefly, the rat (100–200 g) was anesthetized with 4% isoflurane and perfused intracardially with ice-cold oxygenated (95% $O_2$ and 5% $CO_2$) sucrose-substituted artificial cerebrospinal fluid (ACSF) containing (in mM): 252 sucrose, 2.5 KCl, 2 $CaCl_2$, 2 $MgCl_2$, 10 glucose, 26 $NaHCO_3$, 1.25 $NaH_2PO_4$ and 5 kynurenic acid; pH 7.35. The rat was decapitated, the spinal cord was removed by hydraulic extrusion, and 300–400 µm-thick sections were cut from the lumbar enlargement in the parasagittal plane. Slices were kept in normal oxygenated ACSF (126 mM NaCl instead of sucrose and without kynurenic acid) at room temperature until recording.

## Whole-cell patch clamp recordings

Slices were transferred to a recording chamber constantly perfused at ~2 ml/min with oxygenated (95% $O_2$ and 5% $CO_2$) ACSF. Visually identified neurons in lamina I and II were patched in the whole cell configuration with >70% series resistance compensation using an Axopatch 200B amplifier (Molecular Devices). Patch pipettes were pulled from borosilicate glass capillaries (WPI) and were filled with one of two intracellular solutions. For dynamic clamp experiments, the solution comprised (in mM): 125 $KMeSO_4$, 5 KCl, 10 HEPES, 2 $MgCl_2$, 4 ATP, 0.4 GTP as well as 0.1% Lucifer Yellow; pH

was adjusted to 7.2 with KOH. For chloride extrusion measurements, KCl was increased to 25 mM and KMeSO$_4$ was reduced to 115 mM. Recordings were at room temperature (22–26°C). Traces were low-pass filtered at 2 KHz, digitized at 20 KHz using a CED 1401 computer interface (Cambridge Electronic Design), and analyzed offline. Membrane potential (after correction for the liquid junction potential) was adjusted to −65 mV through tonic current injection. Based on responses to series of 500 ms-long current steps, spiking patterns were classified as previously described (*Prescott and De Koninck, 2002*).

## Chloride extrusion measurements

Spiking pattern was determined in current clamp mode immediately after rupturing the membrane. Thereafter, 1 µM tetrodotoxin (Alomone Labs) and 1 µM CGP55845 (Abcam) were bath applied to block sodium and GABA$_B$ channels, respectively. After allowing at least 15 min for the chloride from the pipette solution to dialyze, the cell was voltage clamped at different holding potentials while short (5–10 ms-long) puffs of GABA (1 mM prepared in HEPES-buffered pipette solution) were applied ~5 µm from the soma using a puff pipette (5 MΩ resistance) attached to a picospritzer (Toohey Company, Fairfield, NJ). Experiments were repeated after bath application of 15 µM VU to block KCC2.

## Virtual synaptic input by dynamic clamp

To simulate irregular synaptic input, noisy excitatory and inhibitory conductances ($g_{exc}$ and $g_{inh}$) were constructed from separate Ornstein-Uhlenbeck processes (*Prescott and De Koninck, 2009*; *Uhlenbeck and Ornstein, 1930*) such that $g$ fluctuates randomly while returning to its average value $g_0$ with a time constant τ according to

$$\frac{dg_{exc}}{dt} = -\frac{g_{exc}(t) - g_{exc0}}{\tau_{exc}} + \sqrt{\frac{2}{\tau_{exc}}}\,\sigma_{exc} \cdot \xi_{exc}(0,1),$$

$$\frac{dg_{inh}}{dt} = -\frac{g_{inh}(t) - g_{inh0}}{\tau_{inh}} + \sqrt{\frac{2}{\tau_{inh}}}\,\sigma_{inh} \cdot \xi_{inh}(0,1),$$

where $\xi(0,1)$ is a random number with 0 mean and unit variance and $\sqrt{2/\tau}$ is a scaling factor so that σ specifies the standard deviation of the conductance fluctuations. Based on synaptic decay kinetics, $\tau_{exc}$ = 3 ms and $\tau_{inh}$ = 10 ms give appropriate autocorrelation structure. Conductances were rectified to disallow negative values. The voltage $V$ is recorded from the neuron at each time step, from which the driving forces are calculated and multiplied by the time-varying conductances, and the resultant currents summed according to

$$I_{synaptic}(t) = g_{exc}(t) \cdot [V(t) - E_{exc}] + g_{inh}(t) \cdot [V(t) - E_{inh}]$$

$I_{synaptic}$ was applied to the recorded neuron using the dynamic clamp capability of Signal 5 software (Cambridge Electronic Design). Input-output (*i-o*) curves were plotted by varying $g_{exc0}$ while co-varying $g_{inh0}$ according to a fixed ratio of inhibition to excitation, where α = $g_{inh0}/g_{exc0}$ = 0.5 or two to simulate stimulation in the RF center or surround, respectively. $E_{inh}$ was set to −70 mV or −45 mV to simulate normal and disinhibited conditions, respectively. $E_{exc}$ = 0 mV. Background synaptic activity was blocked with 10 µM bicuculline and 10 µM CNQX (Abcam).

## Statistical analysis

All statistical analysis was conducted in SigmaPlot 11. An appropriate non-parametric test was applied when the normality (Shapiro-Wilk) test failed. Paired tests were used in all cases where responses from the same cell were measured before and after a manipulation. Unpaired tests were used for between-group comparisons. All tests were two-sided. Once a recorded neuron was classified based on its spiking pattern (see Results) and was included or excluded on the basis of that pattern, no data were excluded for any other reason. Sample sizes are evident on the figures, which show data from individual cells, and from the degrees of freedom reported with statistical tests. Sample sizes were not computed because we did not have any prior knowledge of the expected effect size or variance; instead, experiments were continued until we collected >$n$ cells of each type for each experiment, where $n$ = 5 for in vitro experiments and $n$ = 10 for in vivo experiments (because variability is typically greater in vivo). The only exception is in *Figure 2D*, where data collected for analysis in *Figure 1* were re-analyzed after classification described in *Figure 2B*. The total

number or type of cells recorded from each rat is unpredictable, but cells comprising each data set come from ≥5 rats in all cases.

## Acknowledgements

We thank Y De Koninck and MW Salter for their feedback on an early version of this manuscript. This work was supported by a Project Grant from the Canadian Institutes of Health Research (PJT-153161).

## Additional information

### Funding

| Funder | Grant reference number | Author |
| --- | --- | --- |
| Canadian Institutes of Health Research | PJT-153161 | Steven A Prescott |

The funders had no role in study design, data collection and interpretation, or the decision to submit the work for publication.

### Author contributions

Kwan Yeop Lee, Conceptualization, Data curation, Formal analysis, Investigation, Visualization, Methodology, Writing—original draft, Writing—review and editing; Stéphanie Ratté, Formal analysis, Investigation, Writing—review and editing; Steven A Prescott, Conceptualization, Formal analysis, Supervision, Funding acquisition, Visualization, Writing—original draft, Project administration, Writing—review and editing

### Author ORCIDs

Steven A Prescott  https://orcid.org/0000-0002-3827-4512

### Ethics

Animal experimentation: All experimental procedures were approved by the Animal Care Committee at the Hospital for Sick Children (Animal Use Protocol #22919 and #22576) and were performed in accordance to guidelines from the Canadian Council on Animal Care. For in vivo experiments, animals were anesthetized with urethane (20% in normal saline; 1.2 g/kg i.p.). To prepare slices for in vitro experiments, animals were anesthetized with 4% isoflurane.

### Decision letter and Author response

Decision letter https://doi.org/10.7554/eLife.49753.sa1
Author response https://doi.org/10.7554/eLife.49753.sa2

## Additional files

### Supplementary files

• Transparent reporting form

### Data availability

Source data files have been provided for all figures.

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
