## [Decision Letter]

**Acceptance summary:**

Your findings on how excitation and inhibition are physiologically regulated in the spinal dorsal horn and how their imbalance impairs somatosensory processing are very exciting: thank you for choosing *eLife* to report them.

**Decision letter after peer review:**

Thank you for submitting your article "Excitatory neurons are more disinhibited than inhibitory neurons by chloride dysregulation in the spinal dorsal horn" for consideration by *eLife*. Your article has been reviewed by two peer reviewers, and the evaluation has been overseen by Claire Wyart as a Reviewing Editor and Eve Marder as the Senior Editor. The following individual involved in review of your submission has agreed to reveal their identity: Qiufu Ma (Reviewer #2).

The reviewers have discussed the reviews with one another and the Reviewing Editor has drafted this decision to help you prepare a revised submission.

The study by Lee et al. explores how KCC2-mediated chloride dysregulation impacts on putative spinal excitatory neurons (ENs) and inhibitory neurons (INs) located in the dorsal horn, whose identification was based on differential adaption patterns and distinct firing patterns. In depth analysis on how dysfunction of this chloride transporter alters neuronal activity will contribute for a better understanding of the cellular mechanisms behind neuropathic pain. Using a combination of in vivo and in vitro recordings the authors conclude block of KCC2 function has significantly larger effect in excitatory neurons than in inhibitory ones. The authors show first that ENs and INs showed circuit-level difference in E-I balance, with ENs showing stronger excitatory inputs balanced with stronger inhibition, and INs showing weaker excitation balanced with weaker inhibition; as such, disinhibition following KCC2 inhibition leads to larger unmasked excitation in ENs. They also revealed that both ENs and INs show center excitation-surround inhibition, and KCC2 inhibition leads to the recruitment of surround excitatory inputs, which is particularly prominent for ENs; as such, KCC2-mediated chloride dysregulation could lead to a form of spatial summation and subsequent manifestation of dynamic allodynia in response to tactile stimuli across a large skin area.

The question addressed is relevant and the work performed is of high quality. Although the findings are interesting, several important issues need to be addressed in the revised manuscript.

Major comments:

1) In Figure 4 it would be important to investigate if the δ increase in the slope by KCC2 block is different between adapting and non-adapting neurons. In the dynamic clamp experiment, the increase in slope between excitatory in inhibitory neurons was qualitatively replicated but quantitatively it looks likes that the increase in slope was smaller in the dynamic clamp experiment for inhibitory neurons than the one obtained in vivo with KCC2 block. If this is true it needs to be reported and discussed by the authors.

Furthermore, the differential inputs to ENs and INs by rapid-adapting (RA) and slow-adapting (SA) afferents are interesting, and the authors should provide some discussion based on current knowledge of the roles of these afferents in mediating allodynia.

2) Why there is no effect of DHPG in spontaneous firing of adapting cells? Could this be due to the relatively low firing rate of this cells (floor effect)? This is important since it could help understanding why KCC2 block did not impact spontaneous firing on excitatory cells.

3) Why did the authors use a shift in the anion reversal potential from -70 mV to -45mV when the experiment reported in Figure 5 suggest a significantly smaller shift? This needs to be justified/discussed.

4) In the dynamic clamp the difference in spontaneous firing between excitatory and inhibitory cells is relatively small compared to the KCC2 to the data. This needs to be commented.

5) The authors suggest that non-adapting cells receive less excitation as well as less inhibition than adapting neurons (according to the scheme in Figure 8C). However, in Figure 2 the RF area is different between adapting and non-adapting cells. Could this affect the interpretation of Figure 8B? Is it possible that more inhibition in present in the RF center in inhibitory neurons? What was the impact of KCC2 block in the firing of adapting and non-adapting neurons in the center of the RF?

6) The impact of chloride dysregulation on putative ENs and INs was revealed via intrathecal delivery of KCC2 inhibitors or BDNF. While such manipulations allowed the investigators to record the same neurons before and after treatments, they were unable to reveal potential differential KCC2-mediated chloride dysregulation in ENs versus INs that naturally occurred following nerve injury (e.g., only a subset of ENs neurons might have chloride dysregulation). The authors need to point out this caveat in the Discussion.

7) SST has been reported to cause inhibition via activation of sst_2a_ in dynorphin+ inhibitory neurons, which in turn leads to sensitisation of itch transmission ENs. Surprisingly, the authors claimed an activation by SST in non-adapting neurons, which were considered as INs in this study. Further clarification is needed.

8) There are reported conflicts regarding the behavioural outcomes following intrathecal injection of BDNF; the authors mentioned allodynia development in a separate submitted manuscript. During revision and in responses to reviewers' comments, the authors should present the behavioural data in their response to reviewers, particularly the dynamic allodynia that is more relevant for this study, to help to understand the BDNF data described in current manuscript.

---

## [Author Response]

Major comments:1) In Figure 4 it would be important to investigate if the δ increase in the slope by KCC2 block is different between adapting and non-adapting neurons.

For in vivo data in Figure 4B, we now report in the figure legend that the effect of KCC2 blockade does not depend on cell type (i.e. the interaction is not significant; *F*_1,31_ = 0.15, *p* = 0.71) but we acknowledge that larger changes were observed amongst adapting units.

In the dynamic clamp experiment, the increase in slope between excitatory in inhibitory neurons was qualitatively replicated but quantitatively it looks likes that the increase in slope was smaller in the dynamic clamp experiment for inhibitory neurons than the one obtained in vivo with KCC2 block. If this is true it needs to be reported and discussed by the authors.

For dynamic clamp data in Figure 6D, we now report in the figure legend and in the subsection “Excitatory and inhibitory neurons are differentially affected by equivalent chloride dysregulation”, that the effect of shifting *E*_inh_ does not depend on cell type (*F*_1,13_ = 3.81, *p* = 0.073) but the interaction is nearly significant. We point out that larger changes tend to be observed in excitatory neurons.

We write that “the differential increase in firing rate gain between excitatory and inhibitory neurons tends to be greater for in vitro experiments than for in vivoexperiments.” A quantitative comparison between the slope change in dynamic clamp (in vitro) and in vivo experiments is problematic insofar as the units for slope (spk/s/nS for dynamic clamp vs. spk/s/g for von Frey testing) are not the same. The last paragraph of the subsection “Excitatory and inhibitory neurons are differentially affected by equivalent chloride dysregulation”, now highlights the qualitative concordance despite quantitative differences, and incorporates changes made in response to comment #4.

Furthermore, the differential inputs to ENs and INs by rapid-adapting (RA) and slow-adapting (SA) afferents are interesting, and the authors should provide some discussion based on current knowledge of the roles of these afferents in mediating allodynia.

The functional connectivity reported in Figure 2E is now discussed in more depth in the last paragraph of the subsection “Units recorded in vivo comprise two distinct groups”. We explain that the operating mode – coincidence detection vs. integration – may play an important role in dictating how different spinal neurons respond to afferent input. Our results suggest that sustained pressure will engage more inhibition than dynamic touch, which is consistent with gate control theory – you apply pressure rather than tickle a sore area to reduce pain. Our results also relate to efforts to identify which afferent type is sufficient to mediate mechanical allodynia by expressing channelrhodopsin-2 (ChR2) in specific cell types and then selectively photoactivating those cells after nerve injury. The discrepancy between recently published studies suggests that activation of RA but not SA afferents causes allodynia, which would correspond to downstream spinal circuits implementing an NIMPLY logic gate, consistent with our results. This new discussion is included in the aforementioned subsection.

2) Why there is no effect of DHPG in spontaneous firing of adapting cells? Could this be due to the relatively low firing rate of this cells (floor effect)? This is important since it could help understanding why KCC2 block did not impact spontaneous firing on excitatory cells.

We now point out that DHPG does not affect spontaneous spiking (subsection “Adapting and non-adapting units correspond to excitatory and inhibitory neurons, respectively”, second paragraph) and direct readers to the Discussion, where spontaneous spiking and its modulation are discussed at some length (see section entitled “Effects of disinhibition on spontaneous spiking”). We now mention that DHPG did not cause spontaneous firing in excitatory neurons.

We believe that the lack of effect reflects the tendency of adapting units not to spike spontaneously (i.e. in response to slow onset, sustained depolarization) because of their intrinsic excitability rather than a floor effect (see subsection “Effects of disinhibition on spontaneous spiking”, last paragraph). Our Discussion explains that compared with non-adapting units, adapting units do not spike spontaneously after KCC2 blockade, DHPG, or BDNF, despite exhibiting increased evoked spiking under each of those conditions. In our experience with dorsal horn neuron recordings in vitro, tonic-spiking (inhibitory) neurons spike spontaneously if modestly depolarized whereas single- or delayed-spiking (excitatory) neurons do not. Tonic inhibition may also play a role. We believe that our existing discussion does a good job in summarizing and interpreting the spontaneous activity (or lack thereof) under various conditions; directing readers to this discussion when presenting the DHPG data will, we hope, address the reviewers’ concerns and help readers interpret our results.

It is also worth considering that Hu and Gereau, 2011, showed that mGluR5 activation decreases the A-type K current. Because this current activates during abrupt depolarization (as during AMPA receptor-mediated excitation) but inactivates during slow depolarization, the A current preferentially reduces evoked spiking and its downregulation should, therefore, preferentially enhance evoked spiking. This is consistent with more recent work by Zhang et al., 2018, which is now mentioned in the subsection “Adapting and non-adapting units correspond to excitatory and inhibitory neurons, respectively”.

3) Why did the authors use a shift in the anion reversal potential from -70 mV to -45mV when the experiment reported in Figure 5 suggest a significantly smaller shift? This needs to be justified/discussed.

Experiments in Figure 5 were designed to sensitively measure differences in chloride extrusion capacity, not the natural value of *E*_inh_. This is now explained in the figure legend. Applying a high chloride load via the patch pipette reveals how well KCC2 can counteract a chloride load, but that maneuver deliberately disrupts *E*_inh_. Indeed, one tries to “clamp” intracellular chloride concentration and determine how much the measured value of *E*_inh_ deviates from the predicted value because of KCC2-mediated chloride extrusion. Please see Doyon et al., 2016, for a complete explanation. With KCC2 intact, the chloride load causes an incomplete collapse of the chloride gradient, hence the relatively small change in *E*_inh_ pre- vs. post-blockade of KCC2.

For our dynamic clamp experiments, we used published values of *E*_inh_ measured by perforated patch (Coull et al., 2003). As now explained in the subsection “Excitatory and inhibitory neurons are differentially affected by equivalent chloride dysregulation”, we used a value at the top of the published range for simulations since pharmacological blockade of KCC2 strongly reduces chloride extrusion capacity. We now also point out that nerve injury may cause subtler changes in *E*_inh_.

4) In the dynamic clamp the difference in spontaneous firing between excitatory and inhibitory cells is relatively small compared to the KCC2 to the data. This needs to be commented.

Yes, the spontaneous spiking observed in inhibitory neurons is less during dynamic clamp experiments with *E*_inh_ = -45 mV than after KCC2 blockade in vivo. The basis for this difference is unclear but presumably reflects a failure to fully recapitulate in vivo conditions in our in vitro experiments. This is now explained in the last paragraph of the subsection “Excitatory and inhibitory neurons are differentially affected by equivalent chloride dysregulation” alongside changes made in response to comment #1.

5) The authors suggest that non-adapting cells receive less excitation as well as less inhibition than adapting neurons (according to the scheme in Figure 8C). However, in Figure 2 the RF area is different between adapting and non-adapting cells. Could this affect the interpretation of Figure 8B?

This is an important observation and prompted us to improve Figure 8C, and to evaluate it more carefully, leading to the addition of a new panel as Figure 8D (see our last response to point 5). The average RF of non-adapting units is about twice as large as the RF of adapting units at baseline (Figure 2C); The corresponding change in RF diameter is now depicted by the widths of the base of the RFs on the cartoon (see black arrows). The total drive is comparable for each cell type, which is depicted by the total area of the black region.

Is it possible that more inhibition in present in the RF center in inhibitory neurons?

The model in Figure 8C actually predicts (see green arrows) that more inhibition is present in the RF center of excitatory neurons (adapting units). This is now described in the last paragraph of the subsection “Surround inhibition is greater, and disinhibition more consequential, for adapting units”.

What was the impact of KCC2 block in the firing of adapting and non-adapting neurons in the center of the RF?

We tested the prediction identified in (b) by analyzing the response to stimulation of the RF center before and after disinhibition by KCC2 blockade or BDNF. The results, now explained in the subsection “Surround inhibition is greater, and disinhibition more consequential, for adapting units”, revealed that adapting and non-adapting units experienced an equivalent increase in firing. This caused us to re-evaluate our initial model. Indeed, that model also predicted that disinhibition would cause a greater increase in drive from the RF center than from the RF surround, contrary to data presented in Figures 1E and 2D.

We therefore created a revised model, now shown as Figure 8D. In this version, the total drive at baseline is equivalent between adapting and non-adapting units, though the RF is wider for the latter, like in Figure 8C. The crosshatched and hatched areas show the disinhibition-mediated increase in drive from RF center and surround, respectively. The increase in drive from the RF center is equivalent between cell types, but increased drive from the RF surround relative to that from the RF center is about twice as large for adapting units, consistent with Figure 2D. The RF sizes after disinhibition are equal between adapting and non-adapting units, consistent with Figure 2C. The model also depicts the stronger excitatory and inhibitory input to adapting units, consistent with Figure 8B. Overall, we think our revised model does an excellent job accounting for several data sets and we thank the reviewers for posing incisive questions that led us to make these improvements. The improved model is now described in the text and figure legend.

6) The impact of chloride dysregulation on putative ENs and INs was revealed via intrathecal delivery of KCC2 inhibitors or BDNF. While such manipulations allowed the investigators to record the same neurons before and after treatments, they were unable to reveal potential differential KCC2-mediated chloride dysregulation in ENs versus INs that naturally occurred following nerve injury (e.g., only a subset of ENs neurons might have chloride dysregulation). The authors need to point out this caveat in the Discussion.

This issue is addressed at some length in the Results subsection “BDNF induces chloride dysregulation in both cell types”. See also changes made in response to point #3. We have also added a new sentence to the Discussion (subsection “Effects of disinhibition on evoked spiking”) to highlight this issue.

7) SST has been reported to cause inhibition via activation of sst_2a_ in dynorphin+ inhibitory neurons, which in turn leads to sensitisation of itch transmission ENs. Surprisingly, the authors claimed an activation by SST in non-adapting neurons, which were considered as INs in this study. Further clarification is needed.

The preferential effect of SST on non-adapting neurons is consistent with our predictions but the directionality of the effect is opposite to expectations. Specifically, SST has been shown to hyperpolarize sst_2a_-expressing (dynorphin+) neurons whereas we observed an excitatory effect. The reasons for this are not entirely clear, but as now discussed in the third paragraph of the subsection “Adapting and non-adapting units correspond to excitatory and inhibitory neurons, respectively”, the high dose of SST that we applied in vivo may cause receptor internalization, which could in turn reduce normal sst_2a_ receptor signaling. Notably, Kardon et al., 2014, used octreotide rather than SST for in vivo experiments. It is also interesting that the effects of SST reported in early studies were quite variable, probably due to variations in the precise agonist and its concentration.

8) There are reported conflicts regarding the behavioural outcomes following intrathecal injection of BDNF; the authors mentioned allodynia development in a separate submitted manuscript. During revision and in responses to reviewers' comments, the authors should present the behavioural data in their response to reviewers, particularly the dynamic allodynia that is more relevant for this study, to help to understand the BDNF data described in current manuscript.

The effect of intrathecal BDNF on withdrawal threshold (tested by von Frey stimulation) is presented in Figure 4C, D of Mapplebeck et al., 2019. The citation has been updated to reflect the publication of this paper. The collaborator who conducted behavioral testing did not test dynamic allodynia. Indeed, those experiments were conducted prior to publication (Cheng et al., 2017) highlighting the importance of stimulus type. Repeating costly BDNF experiments with brush stimulation was not considered since the type of allodynia was not a focus of our Cell Reports paper.

We are unaware of conflicts regarding the behavioral outcomes of intrathecal BDNF. Several studies have shown that intrathecal BDNF reduces the paw withdrawal threshold (Coull et al., 2005; Ding et al., 2015; Geng et al., 2010; Richner et al., 2019). To our knowledge, the main controversy about BDNF relates to its natural source (microglia vs. primary afferent neurons), which is not critical for our study. If we have not addressed the primary concern, could the reviewers please provide clarification.

Ding, X., Cai, J., Li, S., Liu, X.D., Wan, Y., and Xing, G.G. (2015). BDNF contributes to the development of neuropathic pain by induction of spinal long-term potentiation via SHP2 associated GluN2B-containing NMDA receptors activation in rats with spinal nerve ligation. Neurobiol Dis73, 428-451.

Geng, S.J., Liao, F.F., Dang, W.H., Ding, X., Liu, X.D., Cai, J., Han, J.S., Wan, Y., and Xing, G.G. (2010). Contribution of the spinal cord BDNF to the development of neuropathic pain by activation of the NR2B-containing NMDA receptors in rats with spinal nerve ligation. Exp Neurol222, 256-266.

Richner, M., Pallesen, L.T., Ulrichsen, M., Poulsen, E.T., Holm, T.H., Login, H., Castonguay, A., Lorenzo, L.E., Goncalves, N.P., Andersen, O.M., et al. (2019). Sortilin gates neurotensin and BDNF signaling to control peripheral neuropathic pain. Science advances5, eaav9946.